# AN INEXACT CONDITIONAL GRADIENT METHOD FOR CONSTRAINED BILEVEL OPTIMIZATION

## ABSTRACT

Bilevel optimization is an important class of optimization problems where one optimization problem is nested within another. This framework is widely used in machine learning problems, including meta-learning, data hyper-cleaning, and matrix completion with denoising. In this paper, we focus on a bilevel optimization problem with a strongly convex lower-level problem and a smooth upper-level objective function over a compact and convex constraint set. Several methods have been developed for tackling unconstrained bilevel optimization problems, but there is limited work on methods for the constrained setting. In fact, for those methods that can handle constrained problems, either the convergence rate is slow or the computational cost per iteration is expensive. To address this issue, in this paper, we introduce a novel single-loop projection-free method using a nested approximation technique. Our proposed method has an improved per-iteration complexity, surpassing existing methods, and achieves optimal convergence rate guarantees matching the best-known complexity of projection-free algorithms for solving convex constrained single-level optimization problems. In particular, when the hyper-objective function is convex, our method requires $\tilde{\mathcal{O}}(\epsilon^{-1})$ iterations to find an $\epsilon$-optimal solution. Moreover, when the hyper-objective function is non-convex the complexity of our method is $\mathcal{O}(\epsilon^{-2})$ to find an $\epsilon$-stationary point. We also present numerical experiments to showcase the superior performance of our method compared with state-of-the-art methods.

## 1 INTRODUCTION

Many learning and inference problems take a *hierarchical* form, where one optimization problem is nested within another. Bilevel optimization is often used to model problems of this kind with two levels of hierarchy. In this paper, we consider the bilevel optimization problem of the following form

$$\min_{\mathbf{x} \in \mathcal{X}} \ell(\mathbf{x}) := f(\mathbf{x}, \mathbf{y}^*(\mathbf{x})) \qquad \text{s.t.} \quad \mathbf{y}^*(\mathbf{x}) \in \operatorname*{argmin}_{\mathbf{y} \in \mathbb{R}^m} g(\mathbf{x}, \mathbf{y}). \tag{1}$$

where $n, m \geq 1$ are integers; $\mathcal{X} \subset \mathbb{R}^n$ is a compact and convex set with diameter $D_{\mathcal{X}}$, i.e., $\|\mathbf{x} - \mathbf{y}\| \leq D_{\mathcal{X}}$ for all $\mathbf{x}, \mathbf{y} \in \mathcal{X}$. Further, $f : \mathcal{X} \times \mathbb{R}^m \to \mathbb{R}$ and $g : \mathcal{X} \times \mathbb{R}^m \to \mathbb{R}$ are continuously differentiable functions with respect to (w.r.t.) $\mathbf{x}$ and $\mathbf{y}$ on an open set containing $\mathcal{X}$ and $\mathbb{R}^m$, respectively. Problem (1) involves two optimization problems following a two-level structure. The outer objective $f(\mathbf{x}, \mathbf{y}^*(\mathbf{x}))$ depends on $\mathbf{x}$ both directly and also indirectly through $\mathbf{y}^*(\mathbf{x})$, which is a solution of the lower-level problem of minimizing another function $g$ parameterized by $\mathbf{x}$. Throughout the paper, we assume that $g(\mathbf{x}, \mathbf{y})$ is strongly convex in $\mathbf{y}$, and hence $\mathbf{y}^*(\mathbf{x})$ is uniquely well-defined for all $\mathbf{x} \in \mathcal{X}$. The application of (1) arises in a number of machine learning (ML) problems, such as meta-learning (Rajeswaran et al., 2019), continual learning (Borsos et al., 2020), reinforcement learning (Konda & Tsitsiklis, 1999), hyper-parameter optimization(Franceschi et al., 2018b; Pedregosa, 2016), and data hyper-cleaning (Shaban et al., 2019).

Several methods have been proposed to solve the general form of the bilevel optimization problems mentioned in (1). For instance, using the optimality conditions of the lower-level problem, the works in (Hansen et al., 1992; Shi et al., 2005; Moore, 2010) transformed the bilevel problem into a single-level constrained problem. However, such an approach includes two major challenges: $(i)$ The reduced problem will have too many constraints when the inner problem is large-scale; $(ii)$ Unless the lower-level function $g$ has a specific structure, such as a quadratic form, the optimality condition

of the lower-level problem introduces nonconvexity into the feasible set of the reduced problem. Recently, more efficient gradient-based bilevel optimization algorithms have been proposed, which can be broadly divided into the approximate implicit differentiation (AID) based approach (Pedregosa, 2016; Gould et al., 2016; Domke, 2012; Liao et al., 2018; Ghadimi & Wang, 2018; Lorraine et al., 2020) and the iterative differentiation (ITD) based approach (Shaban et al., 2019; Maclaurin et al., 2015; Franceschi et al., 2018a; Grazzi et al., 2020). Nevertheless, with the exception of a few recent attempts, most of the existing studies have primarily focused on analyzing the asymptotic convergence, leaving room for the development of novel algorithms that come with guaranteed convergence rates.

Furthermore, most prior studies assume $\mathcal{X} = \mathbb{R}^n$, leading to a simpler unconstrained optimization problem. Nonetheless, $\mathcal{X}$ is required to be a strict subset of $\mathbb{R}^n$ in several applications including meta-learning (Franceschi et al., 2018b), personalized federated learning (Fallah et al., 2020), and coreset selection (Borsos et al., 2020). To deal with such constraint sets, one common approach is to use projection-based methods such as projected gradient methods. However, they require solving a nonlinear projection problem on the constraint set and may not be computationally feasible. The limitations of projection-based techniques led to the development of projection-free algorithms like Frank Wolfe-based methods (Frank & Wolfe, 1956). Instead of tackling a non-linear projection problem, as in the case of $\ell_1$-norm or nuclear norm ball constraints, these Frank Wolfe-based techniques need to solve a linear minimization problem over $\mathcal{X}$ with lower computational cost.

In the context of bilevel optimization problems, numerous studies address constrained settings. However, most existing methods primarily rely on projection-based algorithms, with limited exploration of projection-free alternatives. Unfortunately, these methods often exhibit slow convergence rates or high computational costs per iteration. Notably, the rapid convergence rates observed in methods like (Ghadimi & Wang, 2018) are achieved by utilizing the Hessian inverse of the lower-level function, which comes at a steep price, imposing a worst-case computational cost of $\mathcal{O}(m^3)$ and limiting its applicability. To address this issue, an approximation technique for the Hessian inverse was introduced in (Ghadimi & Wang, 2018) and subsequently used in studies such as (Hong et al., 2020; Akhtar et al., 2022). This approximation technique introduces a vanishing bias as the number of inner steps (matrix-vector products) increases, and its computational cost scales with the condition number ($\kappa_g$) of the lower-level problem. Specifically, it incurs a per-iteration complexity of $\mathcal{O}(\kappa_g m^2 \log(K))$.

To overcome these issues, we develop a new inexact projection-free method that achieves optimal convergence rate guarantees for the considered settings while requiring only two matrix-vector products per iteration leading to a complexity of $\mathcal{O}(m^2)$ per iteration. Next, we state our contributions.

**Contributions**. In this paper, we consider a class of bilevel optimization problems with a strongly convex lower-level problem and a smooth upper-level objective function over a compact and convex constraint set. This extends the literature, which has primarily focused on unconstrained problems. We propose a novel single-loop projection-free method that overcomes the limitations of existing approaches by offering improved per-iteration complexity and convergence guarantees. Our main idea is to simultaneously track the trajectories of the lower-level optimal solution as well as the solution to a time-varying quadratic optimization problem. These estimators are calculated using a one-step gradient-type step and are used to estimate the hyper-gradient for a Frank Wolfe-type update. This leads to a scheme that only requires two matrix-vector products at each iteration. Furthermore, existing methods work under the assumption that $\nabla_y f(x, \cdot)$ is uniformly bounded which may not hold in many applications. To address this limitation, we also analyze our proposed method without the gradient boundedness assumption. Our theoretical guarantees for the proposed Inexact Bilevel Conditional Gradient (IBCG) method are as follows:

- When the hyper-objective function $\ell(x)$ is convex, and $\nabla_y f(\mathbf{x}, \cdot)$ is uniformly bounded for any $\mathbf{x} \in \mathcal{X}$, our IBCG method converges in $\tilde{\mathcal{O}}(\kappa_g^4 \epsilon^{-1})$ iterations to reach an $\epsilon$-optimal solution. Relaxing the gradient boundedness assumption to assume Lipschitz continuity results in a convergence rate of $\tilde{\mathcal{O}}(\kappa_g^5 \epsilon^{-1})$.

- When $\ell(x)$ is non-convex and $\nabla_y f(\mathbf{x}, \cdot)$ is bounded, IBCG requires $\mathcal{O}(\kappa_g^4 \epsilon^{-2})$ iterations to find an $\epsilon$-stationary point. This result changes to $\mathcal{O}(\kappa_g^5 \epsilon^{-2})$ when the gradient boundedness assumption is replaced by gradient Lipschitzness.

These results match the best-known complexity of projection-free algorithms for solving convex constrained single-level optimization problems.

Table 1: Summary of results for bilevel optimization with a strongly convex lower-level function. The abbreviations "C", "NC", "PO", "LMO" stand for "convex", "non-convex", "projection oracle", and "linear minimization oracle" respectively and $\kappa_g \triangleq L_g/\mu_g$. $^\dagger$ We use $poly(\kappa_g)$ because authors do not provide the explicit dependency on $\kappa_g$. $^*$ Note that these works focused on convergence rates in the stochastic setting, without addressing the rates in the deterministic setting, therefore, the results are presented for the stochastic setting.

| | Reference | Oracle | Function $\ell$ NC/C | Assumption on $\nabla_y f(\mathbf{x}, \cdot)$ | Overall Complexity | Convergence metric |
|---|---|---|---|---|---|---|
| **Unconstrained** | SUSTAIN* (Khanduri et al., 2021) | — | NC | bounded | $\tilde{\mathcal{O}}(poly(\kappa_g)m^2\epsilon^{-1.5})^\dagger$ | $\mathbb{E}\|\nabla\ell(\mathbf{x}_{k^*})\|^2$ |
| | FSLA* (Li et al., 2022) | — | NC | bounded | $\mathcal{O}(poly(\kappa_g)m^2\epsilon^{-2})$ | $\mathbb{E}\|\nabla\ell(\mathbf{x}_{k^*})\|^2$ |
| | AID-BiO (Ji et al., 2021) | — | NC | bounded | $\mathcal{O}((\kappa_g^{3.5}m^2 + m\kappa_g^4)\epsilon^{-1})$ | $\|\nabla\ell(\mathbf{x}_{k^*})\|^2$ |
| | F$^3$SA (Kwon et al., 2023) | — | NC | bounded | $\tilde{\mathcal{O}}(poly(\kappa_g)m\epsilon^{-1.5})$ | $\|\nabla\ell(\mathbf{x}_{k^*})\|^2$ |
| | RAHGD (Yang et al., 2023) | — | NC | bounded | $\mathcal{O}(\kappa_g^{3.25}m^2\epsilon^{-0.875})$ | $\|\nabla\ell(\mathbf{x}_{k^*})\|^2$ |
| **Constrained** | ABA (Ghadimi & Wang, 2018) | PO | NC | bounded | $\mathcal{O}(\kappa_g^{4.5}m^3\epsilon^{-1} + \kappa_g^5 m\epsilon^{-1.25})$ | $\|\nabla\ell(\mathbf{x}_{k^*})\|^2$ |
| | | | C | | $\mathcal{O}(\kappa_g^{4.5}m^3\epsilon^{-0.5} + \kappa_g^5 m\epsilon^{-0.75})$ | $\ell(\mathbf{x}_{k^*}) - \ell(\mathbf{x}^*)$ |
| | TTSA* (Hong et al., 2020) | PO | NC | bounded | $\tilde{\mathcal{O}}(poly(\kappa_g)m^2\epsilon^{-2.5})$ | $\mathbb{E}\|\mathbf{x}_{k^*} - \text{prox}_{\rho\ell}(\mathbf{x}_{k^*})\|^2$ |
| | | | C | | $\tilde{\mathcal{O}}(poly(\kappa_g)m^2\epsilon^{-4})$ | $\mathbb{E}[\ell(\mathbf{x}_K) - \ell(\mathbf{x}^*)]$ |
| | SBFW* (Akhtar et al., 2022) | LMO | NC | bounded | $\tilde{\mathcal{O}}(poly(\kappa_g)m^2\epsilon^{-4})$ | $\mathbb{E}[\mathcal{G}(\mathbf{x}_{k^*})]$ |
| | | | C | | $\tilde{\mathcal{O}}(poly(\kappa_g)m^2\epsilon^{-3})$ | $\mathbb{E}[\ell(\mathbf{x}_K) - \ell(\mathbf{x}^*)]$ |
| | **Ours** | LMO | NC | Lip. cont. | $\mathcal{O}(\kappa_g^5 m^2\epsilon^{-2})$ | $\mathcal{G}(\mathbf{x}_{k^*})$ |
| | | | C | | $\tilde{\mathcal{O}}(\kappa_g^5 m^2\epsilon^{-1})$ | $\ell(\mathbf{x}_K) - \ell(\mathbf{x}^*)$ |
| | | LMO | NC | bounded | $\mathcal{O}(\kappa_g^4 m^2\epsilon^{-2})$ | $\mathcal{G}(\mathbf{x}_{k^*})$ |
| | | | C | | $\tilde{\mathcal{O}}(\kappa_g^4 m^2\epsilon^{-1})$ | $\ell(\mathbf{x}_K) - \ell(\mathbf{x}^*)$ |

**Related work.** In this section, we review related work on bilevel optimization; also check Table 1 for a summary. Most of the existing works consider unconstrained bilevel problems, i.e., $\mathcal{X} = \mathbb{R}^n$. In particular, (Khanduri et al., 2021; Li et al., 2022) focused on the stochastic setting and proved an overall complexity of $\tilde{\mathcal{O}}(m^2\epsilon^{-1.5})$ and $\mathcal{O}(m^2\epsilon^{-2})$, respectively. In the deterministic setting, (Ji et al., 2020) carefully analyzed the convergence of bilevel algorithms via AID and proved a complexity of $\mathcal{O}((\kappa_g^{3.5}m^2 + m\kappa_g^4)\epsilon^{-1})$, where $\kappa_g$ denotes the condition number of the lower-level objective. By incorporating acceleration techniques, a recent work by (Yang et al., 2023) further improved the complexity to $\mathcal{O}(\kappa_g^{3.25}m^2\epsilon^{-0.875})$. Moreover, to avoid expensive Hessian computation, a fully first-order method is recently proposed by (Kwon et al., 2023) with a complexity of $\tilde{\mathcal{O}}(m\epsilon^{-1.5})$.

In comparison, there are relatively few works on the constrained bilevel optimization problems, which is the considered setting of this paper. (Ghadimi & Wang, 2018) presented an Accelerated Bilevel Approximation (ABA) method consisting of two iterative loops. When the hyper-function is non-convex, it is shown to obtain an overall complexity of $\mathcal{O}(\kappa_g^{4.5}m^3\epsilon^{-1})$ and $\mathcal{O}(\kappa_g^5 m\epsilon^{-1.25})$ in terms of the upper-level and lower-level objective values, respectively. When the hyper-function is convex, the authors further shaved a factor of $\mathcal{O}(\epsilon^{-0.5})$ from the complexities. However, their computational complexity is expensive as they need to compute the Hessian inverse matrix at each iteration, incurring a per-iteration cost of $\mathcal{O}(m^3)$. There have been efforts in designing efficient single-loop methods in order to reduce the per-iteration cost. In particular, similar to our proposed IBCG, the methods in (Dagréou et al., 2022; Li et al., 2022) only require two matrix-vector products per iteration, but these works only considered unconstrained bilevel problems. Built upon the work of (Ghadimi & Wang, 2018), a Two-Timescale Stochastic Approximation (TTSA) algorithm has been proposed (Hong et al., 2020) for constrained bilevel problems in the stochastic setting, which is shown to achieve a complexity of $\tilde{\mathcal{O}}(m^2\epsilon^{-2.5})$ and $\tilde{\mathcal{O}}(m^2\epsilon^{-4})$ when the hyper-function is non-convex and convex, respectively. Concurrently, a penalty-based bilevel gradient descent (PBGD) algorithm was introduced by (Shen & Chen, 2023) in which they analyzed the convergence rate guarantee when the lower-level objective function satisfies the Polyak-Lojasiewicz condition.

It should be noted that the above methods require a projection onto set $\mathcal{X}$ at every iteration. In contrast, our proposed method is projection-free and only requires access to a linear solver, which is suitable for settings where projection is computationally costly; e.g., when $\mathcal{X}$ is a nuclear-norm ball. A closely related work is (Akhtar et al., 2022), where the authors developed a projection-free algorithm (SBFW) for stochastic bilevel optimization problems and it is shown that their method achieves a complexity of $\mathcal{O}(m^2\epsilon^{-4})$ and $\mathcal{O}(m^2\epsilon^{-3})$ for nonconvex and convex settings, respectively.

Finally, we also remark that some concurrent papers consider the case where the lower-level problem can have multiple minima (Liu et al., 2020; Sow et al., 2022; Chen et al., 2023). As they consider a more general setting that brings more challenges, their theoretical results are also weaker, providing only asymptotic convergence guarantees or slower convergence rates.

## 2 PRELIMINARIES

### 2.1 MOTIVATING EXAMPLES

The bilevel optimization formulation in (1) finds applications in various ML problems, including matrix completion (Yokota & Hontani, 2017), meta-learning (Rajeswaran et al., 2019), data hyper-cleaning (Shaban et al., 2019), hyper-parameter optimization (Franceschi et al., 2018b), and more. Next, we delve into two specific examples.

**Matrix Completion with Denoising**: Consider the matrix completion problem, where the objective is to recover missing items from noisy observations of a subset of the matrix's entries. Typically, in noiseless scenarios, the data matrix can be represented as a low-rank matrix, justifying the use of the nuclear norm constraint. However, in applications like image processing and collaborative filtering, noisy observations are common, and relying solely on the nuclear norm constraint can lead to suboptimal results (McRae & Davenport, 2021; Yokota & Hontani, 2017). One approach to incorporate denoising into the matrix completion problem is by formulating it as a bilevel optimization problem (Akhtar et al., 2022), expressed as follows

$$\min_{\|\mathbf{X}\|_* \leq \alpha} \frac{1}{|\Omega_1|} \sum_{(i,j) \in \Omega_1} (\mathbf{X}_{i,j} - \mathbf{Y}_{i,j})^2$$

$$\text{s.t.} \quad \mathbf{Y} \in \operatorname*{argmin}_{\mathbf{V}} \left\{ \frac{1}{|\Omega_2|} \sum_{(i,j) \in \Omega_2} (\mathbf{V}_{i,j} - \mathbf{M}_{i,j})^2 + \lambda_1 \mathcal{R}(\mathbf{V}) + \lambda_2 \|\mathbf{X} - \mathbf{V}\|_F^2 \right\}, \quad (2)$$

where $\mathbf{M} \in \mathbb{R}^{n \times m}$ is the given incomplete noisy matrix and $\Omega$ is the set of observable entries where $\Omega_1$ and $\Omega_2$ represent the set of available entries in upper and lower-level respectively. $\mathcal{R}(\mathbf{V})$ is a regularization term to induce sparsity, e.g., $\ell_1$-norm or pseudo-Huber loss, $\lambda_1$ and $\lambda_2$ are regularization parameters. The presence of the nuclear norm constraint poses a significant challenge in (2). This constraint renders the problem computationally demanding, often making projection-based algorithms impractical. Consequently, there is a compelling need to develop and employ projection-free methods to overcome these computational limitations.

**Model-Agnostic Meta-Learning**: In meta-learning, our aim is to develop models that can adapt effectively to multiple training sets to optimize performance for individual tasks. A widely used formulation for this purpose is model-agnostic meta-learning (MAML) (Finn et al., 2017). MAML seeks to minimize empirical risk across all training sets through an outer objective while using a single step of implicit projected gradient as the inner objective (Rajeswaran et al., 2019). This framework enables efficient model adaptation across various tasks, enhancing performance and flexibility. Consider collections of training and test datasets $\{\mathcal{D}_i^{tr}\}_{i=1}^N$ and $\{\mathcal{D}_i^{test}\}_{i=1}^N$ for $N$ tasks. Implicit MAML can be formulated as a bilevel optimization problem (Rajeswaran et al., 2019)

$$\min_{\boldsymbol{\theta} \in \Theta} \sum_{i=1}^N \ell\left(\mathbf{y}_i^*(\boldsymbol{\theta}), \mathcal{D}_i^{test}\right) \quad \text{s.t.} \quad \mathbf{y}_i^*(\boldsymbol{\theta}) \in \operatorname*{argmin}_{\boldsymbol{\phi}} \left\{ \ell(\boldsymbol{\phi}, \mathcal{D}_i^{tr}) + \frac{\lambda}{2} \|\boldsymbol{\phi} - \boldsymbol{\theta}\|^2 \right\}. \quad (3)$$

Here $\boldsymbol{\theta}$ is the shared model parameter, $\boldsymbol{\phi}$ is the adaptation of $\boldsymbol{\theta}$ to the $i$th training set, and $\ell(\cdot)$ is the loss function. The set $\Theta$ imposes constraints on the model parameter, e.g., $\Theta = \{\boldsymbol{\theta} \mid \|\boldsymbol{\theta}\|_1 \leq r\}$ for some $r > 0$ to induce sparsity. It can be verified that for a sufficiently large value of $\lambda$ the lower-level problem is strongly convex and (3) can be viewed as a special case of (1).

### 2.2 ASSUMPTIONS AND DEFINITIONS

In this subsection, we discuss the definitions and assumptions required throughout the paper. We begin by discussing the assumptions on the upper-level and lower-level objective functions, respectively.

**Assumption 1.** $\nabla_x f(\mathbf{x}, \mathbf{y})$ and $\nabla_y f(\mathbf{x}, \mathbf{y})$ are Lipschitz continuous w.r.t $(\mathbf{x}, \mathbf{y}) \in \mathcal{X} \times \mathbb{R}^m$ such that for any $\mathbf{x}, \bar{\mathbf{x}} \in \mathcal{X}$ and $\mathbf{y}, \bar{\mathbf{y}} \in \mathbb{R}^m$

*(i)* $\|\nabla_x f(\mathbf{x}, \mathbf{y}) - \nabla_x f(\bar{\mathbf{x}}, \bar{\mathbf{y}})\| \leq L_{xx}^f \|\mathbf{x} - \bar{\mathbf{x}}\| + L_{xy}^f \|\bar{\mathbf{y}} - \mathbf{y}\|,$

*(ii)* $\|\nabla_y f(\mathbf{x}, \mathbf{y}) - \nabla_y f(\bar{\mathbf{x}}, \bar{\mathbf{y}})\| \leq L_{yx}^f \|\mathbf{x} - \bar{\mathbf{x}}\| + L_{yy}^f \|\mathbf{y} - \bar{\mathbf{y}}\|.$

**Assumption 2.** $g(\mathbf{x}, \mathbf{y})$ *satisfies the following conditions:*

(i) *For any given* $\mathbf{x} \in \mathcal{X}$, $g(\mathbf{x}, \cdot)$ *is twice continuously differentiable. Moreover,* $\nabla_y g(\cdot, \cdot)$ *is continuously differentiable.*

(ii) *For any* $\mathbf{x} \in \mathcal{X}$, $\nabla_y g(\mathbf{x}, \cdot)$ *is Lipschitz continuous with constant* $L_g \geq 0$. *Moreover, for any* $\mathbf{y} \in \mathbb{R}^m$, $\nabla_y g(\cdot, \mathbf{y})$ *is Lipschitz continuous with constant* $C_{yx}^g \geq 0$.

(iii) *For any* $\mathbf{x} \in \mathcal{X}$, $g(\mathbf{x}, \cdot)$ *is* $\mu_g$-*strongly convex with modulus* $\mu_g > 0$.

(iv) *For any given* $\mathbf{x} \in \mathcal{X}$, $\nabla_{yx}^2 g(\mathbf{x}, \mathbf{y}) \in \mathbb{R}^{n \times m}$ *and* $\nabla_{yy}^2 g(\mathbf{x}, \mathbf{y})$ *are Lipschitz continuous w.r.t* $(\mathbf{x}, \mathbf{y}) \in \mathcal{X} \times \mathbb{R}^m$, *and with constant* $L_{yx}^g \geq 0$ *and* $L_{yy}^g \geq 0$, *respectively.*

**Remark** 2.1. Considering Assumption 2-(ii), we can conclude that $\|\nabla_{yx}^2 g(\mathbf{x}, \mathbf{y})\|$ is bounded with constant $C_{yx}^g \geq 0$ for any $(\mathbf{x}, \mathbf{y}) \in \mathcal{X} \times \mathbb{R}^m$.

To measure the quality of the solution at each iteration, we use the standard Frank-Wolfe gap function associated with the single-level variant of problem (1) formally stated in the next assumption.

**Definition 1** (Convergence Criteria). *When the upper level function* $f(\mathbf{x}, \mathbf{y})$ *is non-convex the Frank-Wolfe gap is defined as*

$$\mathcal{G}(\mathbf{x}) \triangleq \max_{\mathbf{s} \in \mathcal{X}} \{\langle \nabla \ell(\mathbf{x}), \mathbf{x} - \mathbf{s} \rangle\}, \tag{4}$$

*which is a standard performance metric for constrained non-convex settings as mentioned in (Zhang et al., 2020; Reddi et al., 2016). Moreover, in the convex setting, we use the suboptimality gap function, i.e.,* $\ell(\mathbf{x}) - \ell(\mathbf{x}^*)$.

Before proposing our method, we state some important properties related to problem 1 based on the assumptions above:

(I) A standard analysis reveals that given Assumption 2, the optimal solution trajectory of the lower-level problem, i.e., $\mathbf{y}^*(\mathbf{x})$, is Lipschitz continuous (Ghadimi & Wang, 2018).

(II) One of the required properties to develop a method with a convergence guarantee is to show the Lipschitz continuity of the gradient of the single-level objective function. In the literature of bilevel optimization, to show this result it is often required to assume boundedness of $\nabla_y f(\mathbf{x}, \mathbf{y})$ for any $\mathbf{x} \in \mathcal{X}$ and $\mathbf{y} \in \mathbb{R}^m$, e.g., see (Ghadimi & Wang, 2018; Ji et al., 2020; Hong et al., 2020; Akhtar et al., 2022). In contrast, in this paper, we show that this condition is only required for the gradient map $\nabla_y f(\mathbf{x}, \mathbf{y})$ when restricted to the optimal trajectory of the lower-level problem. In particular, we demonstrate that it is sufficient to show the boundedness of $\nabla_y f(\mathbf{x}, \mathbf{y}^*(\mathbf{x}))$ for any $\mathbf{x} \in \mathcal{X}$ which can be proved using the boundedness of constraint set $\mathcal{X}$.

(III) Using the above results we can show that the gradient of the single-level objective function, i.e., $\nabla \ell(\mathbf{x})$, is Lipschitz continuous. This result is one of the main building blocks of the convergence analysis of our proposed method in the next section.

The aforementioned results are formally stated in the following Lemma.

**Lemma 1.** *Suppose Assumptions 1 and 2 hold. Then for any* $\mathbf{x}, \bar{\mathbf{x}} \in \mathcal{X}$, *the following results hold.*

*(I)* $\|\mathbf{y}^*(\mathbf{x}) - \mathbf{y}^*(\bar{\mathbf{x}})\| \leq \mathbf{L_y} \|\mathbf{x} - \bar{\mathbf{x}}\|$, *where* $\mathbf{L_y} \triangleq \frac{C_{yx}^g}{\mu_g}$.

*(II)* $\|\nabla_y f(\mathbf{x}, \mathbf{y}^*(\mathbf{x}))\| \leq C_y^f$, *where* $C_y^f \triangleq \left(L_{yx}^f + \frac{L_{yy}^f C_{yx}^g}{\mu_g}\right) D_{\mathcal{X}} + \|\nabla_y f(\mathbf{x}^*, \mathbf{y}^*(\mathbf{x}^*))\|$.

*(III)* $\|\nabla \ell(\mathbf{x}) - \nabla \ell(\bar{\mathbf{x}})\| \leq \mathbf{L}_\ell \|\mathbf{x} - \bar{\mathbf{x}}\|$, *where* $\mathbf{L}_\ell \triangleq L_{xx}^f + L_{xy}^f \mathbf{L_y} + C_{yx}^g \mathbf{C_v} + \frac{C_y^f}{\mu_g} L_{yx}^g (1 + \mathbf{L_y})$.

## 3 PROPOSED METHOD

As we discussed in section 1, problem (1) can be viewed as a single minimization problem $\min_{\mathbf{x} \in \mathcal{X}} \ell(\mathbf{x})$, however, solving such a problem is a challenging task due to the need for calculation of the lower-level problem's exact solution, a requirement for evaluating the objective function and/or its gradient. In particular, by utilizing Assumptions 1 and 2, it has been shown in (Ghadimi & Wang, 2018) that the gradient of function $\ell(\cdot)$ can be expressed as

$$\nabla \ell(\mathbf{x}) = \nabla_x f(\mathbf{x}, \mathbf{y}^*(\mathbf{x})) - \nabla_{yx}^2 g(\mathbf{x}, \mathbf{y}^*(\mathbf{x})) [\nabla_{yy}^2 g(\mathbf{x}, \mathbf{y}^*(\mathbf{x}))]^{-1} \nabla_y f(\mathbf{x}, \mathbf{y}^*(\mathbf{x})). \tag{5}$$

To implement an iterative method to solve this problem using first-order information, at each iteration $k \geq 0$, one can replace $\mathbf{y}^*(\mathbf{x}_k)$ with an estimated solution $\mathbf{y}_k$ to track the optimal trajectory of the lower-level problem. Such an estimation can be obtained by taking a single gradient descent step with respect to the lower-level objective function. Therefore, the inexact Frank-Wolfe method for the bilevel optimization problem (1) takes the following main steps

---

**Algorithm 1** Inexact Bilevel Conditional Gradient (IBCG) Method

---

1: **Input**: $\{\gamma_k, \eta_k\}_k \subseteq \mathbb{R}_+$, $\alpha > 0$, $\mathbf{x}_0 \in \mathcal{X}$, $\mathbf{y}_0 \in \mathbb{R}^m$
2: **Initialization**: $\mathbf{w}^0 \leftarrow \mathbf{y}^0$
3: **for** $k = 0, \ldots, K - 1$ **do**
4: $\quad \mathbf{w}_{k+1} \leftarrow (I - \eta_k \nabla^2_{yy} g(\mathbf{x}_k, \mathbf{y}_k)) \mathbf{w}_k + \eta_k \nabla_y f(\mathbf{x}_k, \mathbf{y}_k)$
5: $\quad F_k \leftarrow \nabla_x f(\mathbf{x}_k, \mathbf{y}_k) - \nabla^2_{yx} g(\mathbf{x}_k, \mathbf{y}_k) \mathbf{w}_{k+1}$
6: $\quad$ Compute $\mathbf{s}_k \leftarrow \operatorname{argmin}_{\mathbf{s} \in \mathcal{X}} \langle F_k, \mathbf{s} \rangle$
7: $\quad \mathbf{x}_{k+1} \leftarrow (1 - \gamma_k) \mathbf{x}_k + \gamma_k \mathbf{s}_k$
8: $\quad \mathbf{y}_{k+1} \leftarrow \mathbf{y}_k - \alpha \nabla_y g(\mathbf{x}_{k+1}, \mathbf{y}_k)$
9: **end for**

---

$$G_k \leftarrow \nabla_x f(\mathbf{x}_k, \mathbf{y}_k) - \nabla^2_{yx} g(\mathbf{x}_k, \mathbf{y}_k)[\nabla^2_{yy} g(\mathbf{x}_k, \mathbf{y}_k)]^{-1} \nabla_y f(\mathbf{x}_k, \mathbf{y}_k)$$

$$\mathbf{s}_k \leftarrow \operatorname*{argmin}_{\mathbf{s} \in \mathcal{X}} \ \langle G_k, \mathbf{s} \rangle \tag{6a}$$

$$\mathbf{x}_{k+1} \leftarrow (1 - \gamma_k) \mathbf{x}_k + \gamma_k \mathbf{s}_k \tag{6b}$$

$$\mathbf{y}_{k+1} \leftarrow \mathbf{y}_k - \alpha_k \nabla_y g(\mathbf{x}_{k+1}, \mathbf{y}_k). \tag{6c}$$

Calculation of $G_k$ involves Hessian matrix inversion which is computationally costly and requires $\mathcal{O}(m^3)$ operations. To avoid this, one can reformulate the linear minimization subproblem (6a) as

$$\mathbf{s}_k \leftarrow \operatorname*{argmin}_{\mathbf{s} \in \mathcal{X}, \mathbf{d} \in \mathbb{R}^m} \ \langle \nabla_x f(\mathbf{x}_k, \mathbf{y}_k), \mathbf{s} \rangle + \langle \nabla_y f(\mathbf{x}_k, \mathbf{y}_k), \mathbf{d} \rangle$$

$$\text{s.t.} \quad \nabla^2_{yx} g(\mathbf{x}_k, \mathbf{y}_k)^\top \mathbf{s} + \nabla^2_{yy} g(\mathbf{x}_k, \mathbf{y}_k) \mathbf{d} = 0.$$

When the constraint set $\mathcal{X}$ is a polyhedron, the above-reformulated subproblem remains a linear program (LP) with $m$ additional constraints and $m$ additional variables. The resulting LP can be solved using existing algorithms such as interior-point (IP) methods (Karmarkar, 1984). However, there are two primary concerns with this approach. First, if the LMO over $\mathcal{X}$ admits a closed-form solution, the new subproblem may not preserve this structure. Second, when $n = \Omega(m)$, IP methods require at most $\mathcal{O}(m^\omega \log(m/\delta))$ steps at each iteration, where $\delta > 0$ is the desired accuracy and $\mathcal{O}(m^\omega)$ is the time required to multiply two $m \times m$ matrices with $2.37 \leq \omega \leq 3$ (Cohen et al., 2021; van den Brand, 2020). Hence, the computational would be prohibitive for many practical settings highlighting the pressing need for a more efficient algorithm.

### 3.1 MAIN ALGORITHM

As discussed above, there are major limitations in a naive implementation of the FW framework for solving (1) which makes the method in (6) impractical. To propose a practical conditional gradient-based method we revisit the problem's structure. In particular, the gradient of the single-level problem in (5) can be rewritten as follows

$$\nabla \ell(\mathbf{x}) = \nabla_x f(\mathbf{x}, \mathbf{y}^*(\mathbf{x})) - \nabla^2_{yx} g(\mathbf{x}, \mathbf{y}^*(\mathbf{x})) \mathbf{v}(\mathbf{x}), \tag{7a}$$

$$\text{where} \quad \mathbf{v}(\mathbf{x}) \triangleq [\nabla^2_{yy} g(\mathbf{x}, \mathbf{y}^*(\mathbf{x}))]^{-1} \nabla_y f(\mathbf{x}, \mathbf{y}^*(\mathbf{x})). \tag{7b}$$

In this formulation, the effect of Hessian inversion is presented in a separate term $\mathbf{v}(\mathbf{x})$ which can be viewed as the solution of the following *parametric* quadratic programming

$$\mathbf{v}(\mathbf{x}) = \operatorname*{argmin}_{\mathbf{v}} \ \frac{1}{2} \mathbf{v}^\top \nabla^2_{yy} g(\mathbf{x}, \mathbf{y}^*(\mathbf{x})) \mathbf{v} - \nabla_y f(\mathbf{x}, \mathbf{y}^*(\mathbf{x}))^\top \mathbf{v}. \tag{8}$$

Our main idea is to provide *nested* approximations for the true gradient in (5) by estimating trajectories of $\mathbf{y}^*(\mathbf{x})$ and $\mathbf{v}(\mathbf{x})$. To ensure convergence, we carefully control the algorithm's progress in terms of variable $\mathbf{x}$ and limit the error introduced by these approximations. More specifically, at each iteration $k \geq 0$, given an iterate $\mathbf{x}_k$ and an approximated solution of the lower-level problem $\mathbf{y}_k$ we first consider an approximated solution $\tilde{\mathbf{v}}(\mathbf{x}_k)$ of (8) by replacing $\mathbf{y}^*(\mathbf{x}_k)$ with its currently available approximation, i.e., $\mathbf{y}_k$, which leads to the following quadratic programming

$$\tilde{\mathbf{v}}(\mathbf{x}_k) \triangleq \operatorname*{argmin}_{\mathbf{v}} \ \frac{1}{2} \mathbf{v}^\top \nabla^2_{yy} g(\mathbf{x}_k, \mathbf{y}_k) \mathbf{v} - \nabla_y f(\mathbf{x}_k, \mathbf{y}_k)^\top \mathbf{v}.$$

Then $\tilde{\mathbf{v}}(\mathbf{x}_k)$ is approximated with an iterate $\mathbf{w}_{k+1}$ obtained by taking one step of gradient descent with respect to the objective function as follows,

$$\mathbf{w}_{k+1} \leftarrow \mathbf{w}_k - \eta_k \left( \nabla^2_{yy} g(\mathbf{x}_k, \mathbf{y}_k) \mathbf{w}_k - \nabla_y f(\mathbf{x}_k, \mathbf{y}_k) \right),$$

for some step-size $\eta_k \geq 0$. This generates an increasingly accurate sequence $\{\mathbf{w}_k\}_{k\geq 0}$ that tracks the sequence $\{\mathbf{v}(\mathbf{x}_k)\}_{k\geq 0}$. Next, given approximated solutions $\mathbf{y}_k$ and $\mathbf{w}_{k+1}$ for $\mathbf{y}^*(\mathbf{x}_k)$ and $\mathbf{v}(\mathbf{x}_k)$, respectively, we can construct a direction to estimate the hyper-gradient $\nabla\ell(\mathbf{x}_k)$ in (7a). To this end, we construct a direction $F_k = \nabla_x f(\mathbf{x}_k, \mathbf{y}_k) - \nabla^2_{yx} g(\mathbf{x}_k, \mathbf{y}_k)\mathbf{w}_{k+1}$, which determines the next iteration $\mathbf{x}_{k+1}$ using a Frank-Wolfe type update, i.e.,

$$\mathbf{s}_k \leftarrow \operatorname*{argmin}_{\mathbf{s}\in\mathcal{X}}\langle F_k, \mathbf{s}\rangle, \quad \mathbf{x}_{k+1} \leftarrow (1-\gamma_k)\mathbf{x}_k + \gamma_k\mathbf{s}_k.$$

for some step-size $\gamma_k \in [0,1]$. Finally, having an updated decision variable $\mathbf{x}_{k+1}$ we estimate the lower-level optimal solution $\mathbf{y}^*(\mathbf{x}_{k+1})$ by performing another gradient descent step with respect to the lower-level function $g(\mathbf{x}_k, \mathbf{y}_k)$ with step-size $\alpha > 0$ to generate a new iterate $\mathbf{y}_{k+1}$ as follows:

$$\mathbf{y}_{k+1} \leftarrow \mathbf{y}_k - \alpha\nabla_y g(\mathbf{x}_{k+1}, \mathbf{y}_k).$$

Our proposed inexact bilevel conditional gradient (IBCG) method is summarized in Algorithm 1.

To ensure that IBCG has a guaranteed convergence rate, we introduce the following lemma that quantifies the error between the approximated direction $F_k$ from the true direction $\nabla\ell(\mathbf{x}_k)$ at each iteration. This involves providing upper bounds on the errors induced by our nested approximation technique discussed above, i.e., $\|\mathbf{w}_{k+1} - \mathbf{v}(\mathbf{x}_k)\|$ and $\|\mathbf{y}_{k+1} - \mathbf{y}^*(\mathbf{x}_{k+1})\|$, as well as Lemma 1.

**Lemma 2.** *Suppose Assumptions 1-2 hold and let* $\beta \triangleq (L_g - \mu_g)/(L_g + \mu_g)$ *and* $\mathbf{C_v} \triangleq \frac{L^f_{yx} + L^f_{yy}\mathbf{L_y}}{\mu_g} + \frac{C^f_y L^g_{yy}}{\mu^2_g}(1 + \mathbf{L_y})$. *Moreover, let* $\{\mathbf{x}_k, \mathbf{y}_k, \mathbf{w}_k\}_{k\geq 0}$ *be the sequence generated by Algorithm 1 with step-sizes* $\gamma_k = \gamma \in (0,1]$, $\eta_k = \eta < \frac{1-\beta}{\mu_g}$, *and* $\alpha = 2/(\mu_g + L_g)$. *Then, for any* $k \geq 0$

$$\|\nabla\ell(\mathbf{x}_k) - F_k\| \leq \mathbf{C}_2\left(\beta^k D^y_0 + \frac{\gamma\beta\mathbf{L_y}}{1-\beta}D_\mathcal{X}\right) + C^g_{yx}\left(\rho^{k+1}\|\mathbf{w}_0 - \mathbf{v}(\mathbf{x}_0)\| + \frac{\gamma\rho\mathbf{C_v}}{1-\rho}D_\mathcal{X}\right.$$
$$\left. + \frac{\eta\mathbf{C}_1}{\rho-\beta}\rho^{k+2}D^y_0 + \frac{\gamma\beta\mathbf{C}_1\mathbf{L_y}}{(1-\rho)\mu_g}D_\mathcal{X}\right) \tag{9}$$

*where* $\rho \triangleq 1 - \eta\mu_g$, $\mathbf{C}_1 \triangleq L^g_{yy}\frac{C^f_y}{\mu_g} + L^f_{yy}$, $\mathbf{C}_2 \triangleq L^f_{xy} + L^g_{yx}\frac{C^f_y}{\mu_g}$, *and* $D^y_0 \triangleq \|\mathbf{y}_0 - \mathbf{y}^*(\mathbf{x}_0)\|$.

Lemma 2 provides an upper bound on the error of the approximated gradient direction $F_k$. This bound encompasses two types of terms: those that decrease linearly and others that are influenced by the parameter $\gamma$. Selecting the parameter $\gamma$ is a crucial task as larger values can introduce significant errors in the direction taken by the algorithm, while smaller values can impede proper progress in the iterations. Therefore, it is essential to choose $\gamma$ appropriately based on the overall algorithm's progress. By utilizing Lemma 2, we establish a bound on the gap function and ensure a convergence rate guarantee by selecting an appropriate $\gamma$.

## 4 CONVERGENCE ANALYSIS

In this section, we analyze the iteration complexity of our IBCG method. We first consider the case where the objective function of the single-level problem $\ell(\cdot)$ is convex. Before presenting our result, we should mention that there are no generic sufficient and necessary conditions to establish the convexity of $\ell$, and thus it needs to be verified on a case-by-case basis as discussed in (Hong et al., 2020). One sufficient condition for $\ell(\mathbf{x})$ being convex is when $f$ is jointly convex and $y^*(\mathbf{x})$ is linear in $\mathbf{x}$. As another example, consider a min-max optimization problem $\min_{\mathbf{x}\in\mathcal{X}} \max_{\mathbf{y}\in\mathbb{R}^m} f(\mathbf{x}, \mathbf{y})$, where $f$ is convex in $\mathbf{x}$ and strongly-concave in $\mathbf{y}$. This problem can be reformulated as a bilevel optimization problem by letting $g(\mathbf{x}, \mathbf{y}) = -f(\mathbf{x}, \mathbf{y})$, leading to a convex hyper-objective function $\ell(\mathbf{x}) = \max_{\mathbf{y}} f(\mathbf{x}, \mathbf{y})$.

**Theorem 1** (Convex bilevel). *Suppose Assumptions 1 and 2 hold. If* $\ell(\mathbf{x})$ *is convex, let* $\{\mathbf{x}_k\}^{K-1}_{k=0}$ *be the sequence generated by Algorithm 1 with step-sizes specified as in Lemma 2. Then, for all* $K \geq 1$,

$$\ell(\mathbf{x}_K) - \ell(\mathbf{x}^*) \leq (1-\gamma)^K(\ell(\mathbf{x}_0) - \ell(\mathbf{x}^*)) + \sum^{K-1}_{k=0}(1-\gamma)^{K-k}\mathcal{R}_k(\gamma) \tag{10}$$

*where*

$$\mathcal{R}_k(\gamma) \triangleq \gamma\mathbf{C}_2\beta^k D^y_0 D_\mathcal{X} + \frac{\gamma^2\mathbf{C}_2 D^2_\mathcal{X}\mathbf{L_y}\beta}{1-\beta} + C^g_{yx}\Big[\gamma D_\mathcal{X}\rho^{k+1}\|\mathbf{w}_0 - \mathbf{v}(\mathbf{x}_0)\|$$
$$+ \frac{\gamma^2 D^2_\mathcal{X}\rho\mathbf{C_v}}{1-\rho} + \frac{\gamma D_\mathcal{X} D^y_0 \mathbf{C}_1\eta\rho^{k+2}}{\rho-\beta} + \frac{\gamma^2 D^2_\mathcal{X}\mathbf{L_y}\mathbf{C}_1\beta\eta}{(1-\beta)(1-\rho)}\Big] + \frac{1}{2}\mathbf{L}_\ell\gamma^2 D^2_\mathcal{X}. \tag{11}$$

Theorem 1 demonstrates that the suboptimality can be reduced using the upper bound presented in (10), which consists of two components. The first component decreases linearly, while the second term arises from errors in nested approximations and can be mitigated by reducing the step-size $\gamma$. Thus, by carefully selecting the step-size $\gamma$, we can achieve a guaranteed convergence rate as outlined in the following Corollary. In particular, we establish that setting $\gamma = \log(K)/K$ yields a convergence rate of $\mathcal{O}(\log(K)/K)$.

**Corollary 1.** *Let $\{\mathbf{x}_k\}_{k=0}^{K-1}$ be the sequence generated by Algorithm 1 with step-size $\gamma_k = \gamma = \frac{\log(K)}{K}$. Under the premises of Theorem 1 we have that $\ell(\mathbf{x}_K) - \ell(\mathbf{x}^*) \leq \epsilon$ after $\mathcal{O}(\kappa_g^5 \epsilon^{-1} \log(\epsilon^{-1}))$ iterations. Furthermore, assuming that $\nabla_y f(\mathbf{x}, \cdot)$ is uniformly bounded for any $\mathbf{x} \in \mathcal{X}$, we have that $\ell(\mathbf{x}_k) - \ell(\mathbf{x}^*) \leq \epsilon$ after $\mathcal{O}(\kappa_g^4 \epsilon^{-1} \log(\epsilon^{-1}))$ iterations.*

Now we turn to the case where the objective function of the single-level problem $\ell(\cdot)$ is non-convex.

**Theorem 2** (Non-convex bilevel). *Suppose that Assumption 1 and 2 hold. Let $\{\mathbf{x}_k\}_{k=0}^{K-1}$ be the sequence generated by Algorithm 1 with step-sizes specified as in Lemma 2. Then,*

$$
\begin{aligned}
\mathcal{G}_{k^*} \leq{}& \frac{\ell(\mathbf{x}_0) - \ell(\mathbf{x}^*)}{K\gamma} + \frac{\gamma \mathbf{C}_2 D_\mathcal{X} \mathbf{L_y} \beta}{1-\beta} + \frac{\gamma D_\mathcal{X}^2 \rho \mathbf{C_v} C_{yx}^g \rho}{1-\rho} + \frac{\gamma D_\mathcal{X}^2 C_{yx}^g \mathbf{L_y} \mathbf{C}_1 \beta \eta}{(1-\beta)(1-\rho)} + \frac{1}{2} \mathbf{L}_\ell \gamma D_\mathcal{X}^2 \\
&+ \frac{\mathbf{C}_2 D_0^y D_\mathcal{X} \beta}{K(1-\beta)} + \frac{D_\mathcal{X} C_{yx}^g \rho \|\mathbf{w}_0 - \mathbf{v}(\mathbf{x}_0)\|}{K(1-\rho)} + \frac{D_\mathcal{X} D_0^y C_{yx}^g \mathbf{C}_1 \eta \rho^2}{K(1-\beta)(1-\rho)}
\end{aligned}
\tag{12}
$$

*where $\mathcal{G}_{k^*}$ is defined as $\mathcal{G}_{k^*} \triangleq \min_{0 \leq k \leq K-1} \mathcal{G}(\mathbf{x}_k)$.*

Theorem 2 establishes an upper bound on the Frank-Wolfe gap for the iterates generated by IBCG. It shows that the Frank-Wolfe gap vanishes when the step-size $\gamma$ is properly selected. Specifically, setting $\gamma = \mathcal{O}(1/\sqrt{K})$ as outlined in the next Corollary results in a convergence rate of $\mathcal{O}(1/\sqrt{K})$.

**Corollary 2.** *Let $\{\mathbf{x}_k\}_{k=0}^{K-1}$ be the sequence generated by Algorithm 1 with step-size $\gamma_k = \gamma = \kappa_g^{-2.5} K^{-0.5}$, then there exists $k^* \in \{0, 1, \ldots, K-1\}$ such that $\mathcal{G}_{k^*} \leq \epsilon$ after $\mathcal{O}(\kappa_g^5 \epsilon^{-2})$ iterations. Furthermore, assuming that $\nabla_y f(\mathbf{x}, \cdot)$ is uniformly bounded for any $\mathbf{x} \in \mathcal{X}$, selecting $\gamma_k = \gamma = \kappa_g^{-2} K^{-0.5}$ implies that $\mathcal{G}_{k^*} \leq \epsilon$ after $\mathcal{O}(\kappa_g^4 \epsilon^{-2})$ iterations.*

**Remark** 4.1. It is worth emphasizing that our proposed method requires only two matrix-vector multiplications, which significantly contributes to its efficiency. Furthermore, our results represent the state-of-the-art bound for the considered setting, with a near-optimal complexity among projection-free methods for single-level optimization problems. This is noteworthy as it is known that the worst-case complexity of such methods is $\Theta(1/\epsilon)$ (Jaggi, 2013; Lan, 2013). Our complexity result in the non-convex setting also matches the best-known bound of $\Theta(1/\epsilon^2)$ within the family of projection-free methods for single-level optimization problems (Jaggi, 2013; Lan, 2013). This optimality underscores the efficiency and effectiveness of our approach in this particular context.

## 5 NUMERICAL EXPERIMENTS

In this section, we test our method for solving different bilevel optimization problems. We consider the matrix completion with the denoising example described in Section 2 and compare our method with methods proposed in (Hong et al., 2020; Akhtar et al., 2022). All the experiments are performed in MATLAB R2022a with Intel(R) Core(TM) i5-10210U CPU @ 1.60GHz. Further numerical experiments are presented in Appendix G.

### 5.1 MATRIX COMPLETION WITH DENOISING

In this section, we study the performance of our proposed IBCG algorithm for solving matrix completion with denoising problem in (2) for both synthetic and real dataset.

**Synthetic dataset.** The experimental setup we adopt is aligned with the methodology used in (Mokhtari et al., 2020). In particular, we create an observation matrix $M = \hat{X} + E$. In this setting $\hat{X} = WW^T$ where $W \in \mathbb{R}^{n \times r}$ containing normally distributed independent entries, and $E = \hat{n}(L + L^T)$ is a noise matrix where $L \in \mathbb{R}^{n \times n}$ containing normally distributed independent entries and $\hat{n} \in (0, 1)$ is the noise factor. During the simulation process, we set $n = 250$, $r = 10$,

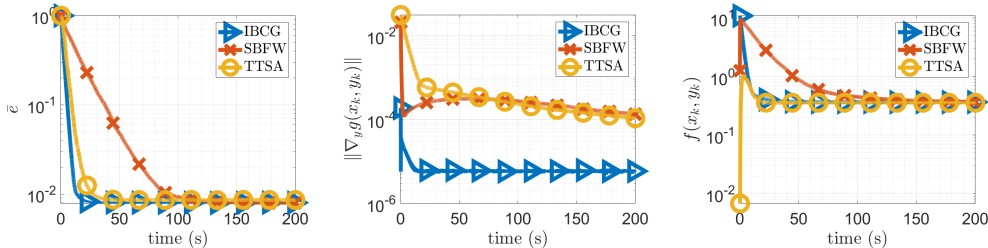

Figure 1: Performance of IBCG vs SBFW and TTSA on problem (2) for synthetic dataset. Plots from left to right: normalized error ($\bar{e}$), $\|\nabla_y g(x_k, y_k)\|$, and $f(x_k, y_k)$ over time.

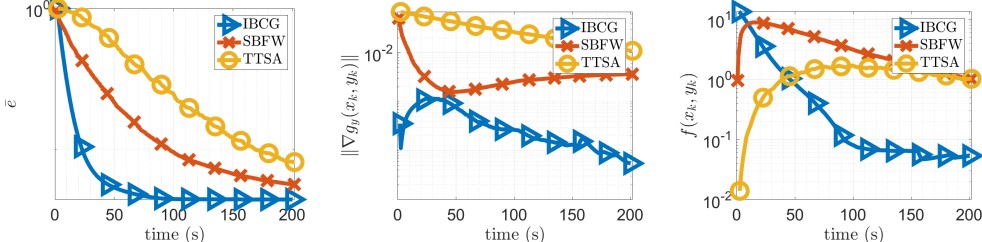

Figure 2: The performance of IBCG (blue) vs SBFW (red) and TTSA (yellow) on problem (2) for real dataset. Plots from left to right: normalized error ($\bar{e}$), $\|\nabla_y g(x_k, y_k)\|$, and $f(x_k, y_k)$ over time.

and $\alpha = \|\hat{X}\|_*$. Additionally, we establish the set of observed entries $\Omega$ by randomly sampling $M$ entries with a probability of 0.8. Initially, we set $\hat{n}$ to be 0.5 and employ the IBCG algorithm to solve the problem described in equation (2). To evaluate the performance of our proposed method, we compare it with state-of-the-art methods in the literature for constrained bilevel optimization problems, specifically TTSA (Hong et al., 2020) and SBFW (Akhtar et al., 2022). We set $\lambda_1 = \lambda_2 = 0.05$, and set the maximum number of iteration as $10^4$. It should be noted that we consider pseudo-Huber loss defined by $\mathcal{R}_\delta(\mathbf{V}) = \sum_{i,j} \delta^2(\sqrt{1 + (\mathbf{V}_{ij}/\delta)^2} - 1)$ as a regularization term to induce sparsity and set $\delta = 0.9$. The performance is analyzed based on the normalized error, defined as $\bar{e} = \sum_{(i,j) \in \Omega}(X_{i,j} - \hat{X}_{i,j})^2 / \sum_{(i,j) \in \Omega}(\hat{X}_{i,j})^2$, where $X$ is the matrix generated by the algorithm. We note that SBFW algorithm suffers from a slower theoretical convergence rate compared to projection-based schemes, but our proposed method outperforms other algorithms by achieving lower values of $\|\nabla_y g(x_k, y_k)\|$ and slightly better performance in terms of the normalized error values – see Figure 1. This gain comes from the projection-free nature of the proposed algorithm and its fast convergence since we are no longer required to perform a complicated projection at each iteration.

**Real dataset.** To evaluate the scalability of IBCG, an experiment was done using the MovieLens datasets, containing large-size matrices. The datasets consist of user-generated movie ratings, ranging from 1 to 5. First, we utilized the MovieLens 100k dataset, which consists of $10^5$ ratings collected from a sample of 1000 individuals encompassing a selection of 1700 movies. This dataset is represented by the observation matrix $M \in \mathbb{R}^{1000 \times 1700}$. Figure 2 shows the performance of different methods and we observe that our proposed method achieves a faster convergence in terms of the normalized error $\bar{e}$, lower-level optimality $\|\nabla_y g(x_k, y_k)\|$, and upper-level objective function value $f(x_k, y_k)$. In order to emphasize the practical significance of the projection-free bilevel approach, we conducted additional experiments on a more extensive dataset. The results are in Appendix G.3.

## 6   CONCLUSION

In this paper, we focused on the constrained bilevel optimization problem that has a wide range of applications in learning problems. We proposed a novel single-loop projection-free method based on nested approximation techniques, which offers optimal convergence rate guarantees that match the best-known complexity of projection-free algorithms for solving convex constrained single-level optimization problems. In particular, we proved that our proposed method requires approximately $\tilde{\mathcal{O}}(\epsilon^{-1})$ iterations to find an $\epsilon$-optimal solution when the hyper-objective function $\ell(x)$ is convex, and approximately $\mathcal{O}(\epsilon^{-2})$ to find an $\epsilon$-stationary point when $\ell(x)$ is non-convex. Our numerical results also showed superior performance of our IBCG algorithm compared to existing algorithms.

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

APPENDIX

In section A, we establish technical lemmas based on the assumptions considered in the paper. These lemmas characterize important properties of problem 1. Notably, Lemma 1 is instrumental in understanding the properties associated with problem 1. Moving on to section B, we present a series of lemmas essential for deriving the rate results of the proposed algorithm. Among them, Lemma 2 quantifies the error between the approximated direction $F_k$ and $\nabla \ell(\mathbf{x}_k)$. This quantification plays a crucial role in establishing the one-step improvement lemma (see Lemma 7). Next, we provide the proofs of Theorem 1 and Corollary 1 in sections C and D, respectively, that support the results presented in the paper for the convex scenario. Finally, in sections E and F we provide the proofs for Theorem 2 along with Corollary 2 for the nonconvex scenario.

## A  SUPPORTING LEMMAS

In this section, we provide detailed explanations and proofs for the lemmas supporting the main results of the paper.

### A.1  PROOF OF LEMMA 1

**(I)** Recall that $\mathbf{y}^*(\mathbf{x})$ is the minimizer of the lower-level problem whose objective function is strongly convex, therefore,

$$\mu_g \left\| \mathbf{y}^*(\mathbf{x}) - \mathbf{y}^*(\bar{\mathbf{x}}) \right\|^2 \leq \langle \nabla_y g(\mathbf{x}, \mathbf{y}^*(\mathbf{x})) - \nabla_y g(\mathbf{x}, \mathbf{y}^*(\bar{\mathbf{x}})), \mathbf{y}^*(\mathbf{x}) - \mathbf{y}^*(\bar{\mathbf{x}}) \rangle$$
$$= \langle \nabla_y g(\bar{\mathbf{x}}, \mathbf{y}^*(\bar{\mathbf{x}})) - \nabla_y g(\mathbf{x}, \mathbf{y}^*(\bar{\mathbf{x}})), \mathbf{y}^*(\mathbf{x}) - \mathbf{y}^*(\bar{\mathbf{x}}) \rangle$$

Note that $\nabla_y g(\mathbf{x}, \mathbf{y}^*(\mathbf{x})) = \nabla_y g(\bar{\mathbf{x}}, \mathbf{y}^*(\bar{\mathbf{x}})) = 0$. Using the Cauchy-Schwartz inequality we have

$$\mu_g \left\| \mathbf{y}^*(\mathbf{x}) - \mathbf{y}^*(\bar{\mathbf{x}}) \right\|^2 \leq \|\nabla_y g(\bar{\mathbf{x}}, \mathbf{y}^*(\bar{\mathbf{x}})) - \nabla_y g(\mathbf{x}, \mathbf{y}^*(\bar{\mathbf{x}}))\| \|\mathbf{y}^*(\mathbf{x}) - \mathbf{y}^*(\bar{\mathbf{x}})\|$$
$$\leq C_{yx}^g \|\mathbf{x} - \bar{\mathbf{x}}\| \|\mathbf{y}^*(\mathbf{x}) - \mathbf{y}^*(\bar{\mathbf{x}})\|$$

where the last inequality is obtained by using the Assumption 2. Therefore, we conclude that $\mu_g \left\| \mathbf{y}^*(\mathbf{x}) - \mathbf{y}^*(\bar{\mathbf{x}}) \right\| \leq C_{yx}^g \|\mathbf{x} - \bar{\mathbf{x}}\|$ which leads to the desired result in part (I).

**(II)** We first show that the function $\mathbf{x} \mapsto \nabla_y f(\mathbf{x}, \mathbf{y}^*(\mathbf{x}))$ is Lipschitz continuous. To see this, note that for any $\mathbf{x}, \bar{\mathbf{x}} \in \mathcal{X}$, we have

$$\|\nabla_y f(\mathbf{x}, \mathbf{y}^*(\mathbf{x})) - \nabla_y f(\bar{\mathbf{x}}, \mathbf{y}^*(\bar{\mathbf{x}}))\| \leq L_{yx}^f \|\mathbf{x} - \bar{\mathbf{x}}\| + L_{yy}^f \|\mathbf{y}^*(\mathbf{x}) - \mathbf{y}^*(\bar{\mathbf{x}})\|$$
$$\leq \left( L_{yx}^f + \frac{L_{yy}^f C_{yx}^g}{\mu_g} \right) \|\mathbf{x} - \bar{\mathbf{x}}\|,$$

where in the last inequality we used Lemma 1-(I). Since $\mathcal{X}$ is bounded, we also have $\|\mathbf{x} - \bar{\mathbf{x}}\| \leq D_{\mathcal{X}}$. Therefore, letting $\bar{\mathbf{x}} = \mathbf{x}^*$ in the above inequality and using the triangle inequality, we have

$$\|\nabla_y f(\mathbf{x}, \mathbf{y}^*(\mathbf{x}))\| \leq \left( L_{yx}^f + \frac{L_{yy}^f C_{yx}^g}{\mu_g} \right) D_{\mathcal{X}} + \|\nabla_y f(\mathbf{x}^*, \mathbf{y}^*(\mathbf{x}^*))\|.$$

Thus, we complete the proof by letting $C_y^f = \left( L_{yx}^f + \frac{L_{yy}^f C_{yx}^g}{\mu_g} \right) D_{\mathcal{X}} + \|\nabla_y f(\mathbf{x}^*, \mathbf{y}^*(\mathbf{x}^*))\|$.

Before proceeding to show the result of part (III) of Lemma 1, we first establish an auxiliary lemma stated next.

**Lemma 3.** *Under the premises of Lemma 1, we have that for any* $\mathbf{x}, \bar{\mathbf{x}} \in \mathcal{X}$, $\|\mathbf{v}(\mathbf{x}) - \mathbf{v}(\bar{\mathbf{x}})\| \leq \mathbf{C_v} \|\mathbf{x} - \bar{\mathbf{x}}\|$ *for some* $\mathbf{C_v} \geq 0$.

*Proof.* We start the proof by recalling that $\mathbf{v}(\mathbf{x}) = \nabla_{yy}^2 g(\mathbf{x}, \mathbf{y}^*(\mathbf{x}))^{-1} \nabla_y f(\mathbf{x}, \mathbf{y}^*(\mathbf{x}))$. Next, adding and subtracting $\nabla_{yy}^2 g(\mathbf{x}, \mathbf{y}^*(\mathbf{x})) \nabla_y f(\bar{\mathbf{x}}, \mathbf{y}^*(\bar{\mathbf{x}}))$ followed by a triangle inequality leads to,

$$\|\mathbf{v}(\mathbf{x}) - \mathbf{v}(\bar{\mathbf{x}})\|$$
$$= \|[\nabla_{yy}^2 g(\mathbf{x}, \mathbf{y}^*(\mathbf{x}))]^{-1} \nabla_y f(\mathbf{x}, \mathbf{y}^*(\mathbf{x})) - [\nabla_{yy}^2 g(\bar{\mathbf{x}}, \mathbf{y}^*(\bar{\mathbf{x}}))]^{-1} \nabla_y f(\bar{\mathbf{x}}, \mathbf{y}^*(\bar{\mathbf{x}}))\|$$
$$\leq \|[\nabla_{yy}^2 g(\mathbf{x}, \mathbf{y}^*(\mathbf{x}))]^{-1} \big(\nabla_y f(\mathbf{x}, \mathbf{y}^*(\mathbf{x})) - \nabla_y f(\bar{\mathbf{x}}, \mathbf{y}^*(\bar{\mathbf{x}}))\big)\| + \|\big([\nabla_{yy}^2 g(\mathbf{x}, \mathbf{y}^*(\mathbf{x}))]^{-1}$$
$$- [\nabla_{yy}^2 g(\bar{\mathbf{x}}, \mathbf{y}^*(\bar{\mathbf{x}}))]^{-1}\big) \nabla_y f(\bar{\mathbf{x}}, \mathbf{y}^*(\bar{\mathbf{x}}))\|$$
$$\leq \frac{1}{\mu_g} \big(L_{yx}^f \|\mathbf{x} - \bar{\mathbf{x}}\| + L_{yy}^f \|\mathbf{y}^*(\mathbf{x}) - \mathbf{y}^*(\bar{\mathbf{x}})\|\big) + C_y^f \|[\nabla_{yy}^2 g(\mathbf{x}, \mathbf{y}^*(\mathbf{x}))]^{-1} - [\nabla_{yy}^2 g(\bar{\mathbf{x}}, \mathbf{y}^*(\bar{\mathbf{x}}))]^{-1}\|,$$
$$\tag{13}$$

where in the last inequality we used Assumptions 1 and 2-(iii) along with the premises of Lemma 1-(II). Moreover, for any invertible matrices $H_1$ and $H_2$, we have that

$$\|H_2^{-1} - H_1^{-1}\| = \|H_1^{-1}\big(H_1 - H_2\big) H_2^{-1}\| \leq \|H_1^{-1}\| \|H_2^{-1}\| \|H_1 - H_2\|. \tag{14}$$

Therefore, using the result of Lemma 1-(I) and (14) we can further bound inequality (13) as follows,

$$\|\mathbf{v}(\mathbf{x}) - \mathbf{v}(\bar{\mathbf{x}})\|$$
$$\leq \frac{1}{\mu_g} \big(L_{yx}^f \|\mathbf{x} - \bar{\mathbf{x}}\| + L_{yy}^f \mathbf{L_y} \|\mathbf{x} - \bar{\mathbf{x}}\|\big) + C_y^f \|[\nabla_{yy}^2 g(\mathbf{x}, \mathbf{y}^*(\mathbf{x}))]^{-1} - [\nabla_{yy}^2 g(\bar{\mathbf{x}}, \mathbf{y}^*(\bar{\mathbf{x}}))]^{-1}\|$$
$$\leq \frac{1}{\mu_g} \big(L_{yx}^f + L_{yy}^f \mathbf{L_y}\big) \|\mathbf{x} - \bar{\mathbf{x}}\| + \frac{C_y^f}{\mu_g^2} L_{yy}^g \big(\|\mathbf{x} - \bar{\mathbf{x}}\| + \|\mathbf{y}^*(\mathbf{x}) - \mathbf{y}^*(\bar{\mathbf{x}})\|\big)$$
$$= \Big(\frac{L_{yx}^f + L_{yy}^f \mathbf{L_y}}{\mu_g} + \frac{C_y^f L_{yy}^g}{\mu_g^2}(1 + \mathbf{L_y})\Big) \|\mathbf{x} - \bar{\mathbf{x}}\|.$$

The result follows by letting $\mathbf{C_v} = \frac{L_{yx}^f + L_{yy}^f \mathbf{L_y}}{\mu_g} + \frac{C_y^f L_{yy}^g}{\mu_g^2}(1 + \mathbf{L_y})$. $\qquad\square$

**(III)** We start proving this part using the definition of $\nabla \ell(\mathbf{x})$ stated in (7a). Utilizing the triangle inequality we obtain

$$\|\nabla \ell(\mathbf{x}) - \nabla \ell(\bar{\mathbf{x}})\|$$
$$= \|\nabla_x f(\mathbf{x}, \mathbf{y}^*(\mathbf{x})) - \nabla_{yx}^2 g(\mathbf{x}, \mathbf{y}^*(\mathbf{x})) \mathbf{v}(\mathbf{x}) - \big(\nabla_x f(\bar{\mathbf{x}}, \mathbf{y}^*(\bar{\mathbf{x}})) - \nabla_{yx}^2 g(\bar{\mathbf{x}}, \mathbf{y}^*(\bar{\mathbf{x}})) \mathbf{v}(\bar{\mathbf{x}})\big)\|$$
$$\leq \|\nabla_x f(\mathbf{x}, \mathbf{y}^*(\mathbf{x})) - \nabla_x f(\bar{\mathbf{x}}, \mathbf{y}^*(\bar{\mathbf{x}}))\| + \|\big[\nabla_{yx}^2 g(\bar{\mathbf{x}}, \mathbf{y}^*(\bar{\mathbf{x}})) \mathbf{v}(\bar{\mathbf{x}}) - \nabla_{yx}^2 g(\bar{\mathbf{x}}, \mathbf{y}^*(\bar{\mathbf{x}})) \mathbf{v}(\mathbf{x})\big]$$
$$+ \big[\nabla_{yx}^2 g(\bar{\mathbf{x}}, \mathbf{y}^*(\bar{\mathbf{x}})) \mathbf{v}(\mathbf{x}) - \nabla_{yx}^2 g(\mathbf{x}, \mathbf{y}^*(\mathbf{x})) \mathbf{v}(\mathbf{x})\big]\| \tag{15}$$

where the second term of the RHS follows from adding and subtracting the term $\nabla_{yx}^2 g(\bar{\mathbf{x}}, \mathbf{y}^*(\bar{\mathbf{x}})) \mathbf{v}(\mathbf{x})$. Next, from Assumptions 1-(i) and 2-(v) together with the triangle inequality application we conclude that

$$\|\nabla \ell(\mathbf{x}) - \nabla \ell(\bar{\mathbf{x}})\| \leq L_{xx}^f \|\mathbf{x} - \bar{\mathbf{x}}\| + L_{xy}^f \|\mathbf{y}^*(\mathbf{x}) - \mathbf{y}^*(\bar{\mathbf{x}})\| + C_{yx}^g \|\mathbf{v}(\bar{\mathbf{x}}) - \mathbf{v}(\mathbf{x})\|$$
$$+ \frac{C_y^f}{\mu_g} \|\nabla_{yx}^2 g(\bar{\mathbf{x}}, \mathbf{y}^*(\bar{\mathbf{x}})) - \nabla_{yx}^2 g(\mathbf{x}, \mathbf{y}^*(\mathbf{x}))\| \tag{16}$$

It should be that in the last inequality, we use the fact that $\|\mathbf{v}(\mathbf{x})\| = \|[\nabla_{yy}^2 g(\mathbf{x}, \mathbf{y}^*(\mathbf{x}))]^{-1} \nabla_y f(\mathbf{x}, \mathbf{y}^*(\mathbf{x}))\| \leq \frac{C_y^f}{\mu_g}$. Combining the result of Lemma 1 part (I) and (II) with the Assumption 2-(iv) leads to

$$\|\nabla \ell(\mathbf{x}) - \nabla \ell(\bar{\mathbf{x}})\| \leq L_{xx}^f \|\mathbf{x} - \bar{\mathbf{x}}\| + L_{xy}^f \mathbf{L_y} \|\mathbf{x} - \bar{\mathbf{x}}\| + C_{yx}^g \mathbf{C_v} \|\mathbf{x} - \bar{\mathbf{x}}\|$$
$$+ \frac{C_y^f}{\mu_g} L_{yx}^g \big(\|\mathbf{x} - \bar{\mathbf{x}}\| + \|\mathbf{y}^*(\mathbf{x}) - \mathbf{y}^*(\bar{\mathbf{x}})\|\big)$$
$$\leq L_{xx}^f \|\mathbf{x} - \bar{\mathbf{x}}\| + L_{xy}^f \mathbf{L_y} \|\mathbf{x} - \bar{\mathbf{x}}\| + C_{yx}^g \mathbf{C_v} \|\mathbf{x} - \bar{\mathbf{x}}\|$$
$$+ \frac{C_y^f}{\mu_g} L_{yx}^g \big(\|\mathbf{x} - \bar{\mathbf{x}}\| + \mathbf{L_y} \|\mathbf{x} - \bar{\mathbf{x}}\|\big)$$
$$\leq \Big(L_{xx}^f + L_{xy}^f \mathbf{L_y} + C_{yx}^g \mathbf{C_v} + \frac{C_y^f}{\mu_g} L_{yx}^g (1 + \mathbf{L_y})\Big) \|\mathbf{x} - \bar{\mathbf{x}}\| \tag{17}$$

The desired result can be obtained by letting $\mathbf{L}_\ell = L_{xx}^f + L_{xy}^f \mathbf{L_y} + C_{yx}^g \mathbf{C_v} + \frac{C_y^f}{\mu_g} L_{yx}^g (1 + \mathbf{L_y})$. $\qquad \square$

## B  REQUIRED LEMMAS FOR THEOREMS 1 AND 2

Before we proceed to the proofs of Theorems 1 and 2, we present the following technical lemmas which quantify the error between the approximated solution $\mathbf{y}_k$ and $\mathbf{y}^*(\mathbf{x}_k)$, as well as between $\mathbf{w}_{k+1}$ and $\mathbf{v}(\mathbf{x}_k)$.

**Lemma 4.** *Suppose Assumption 2 holds. Let $\{(\mathbf{x}_k, \mathbf{y}_k)\}_{k \geq 0}$ be the sequence generated by Algorithm 1, such that $\alpha = 2/(\mu_g + L_g)$. Then, for any $k \geq 0$*

$$\|\mathbf{y}_k - \mathbf{y}^*(\mathbf{x}_k)\| \leq \beta^k \|\mathbf{y}_0 - \mathbf{y}^*(\mathbf{x}_0)\| + \mathbf{L_y} D_\mathcal{X} \sum_{i=0}^{k-1} \gamma_i \beta^{k-i}, \tag{18}$$

*where $\beta \triangleq (L_g - \mu_g)/(L_g + \mu_g)$.*

*Proof.* We begin the proof by characterizing the one-step progress of the lower-level iterate sequence $\{\mathbf{y}_k\}_k$. Indeed, at iteration $k$ we aim to approximate $\mathbf{y}^*(\mathbf{x}_{k+1}) = \operatorname{argmin}_\mathbf{y} g(\mathbf{x}_{k+1}, \mathbf{y})$. According to the update of $\mathbf{y}_{k+1}$ we observe that

$$\begin{aligned}
\|\mathbf{y}_{k+1} - \mathbf{y}^*(\mathbf{x}_{k+1})\|^2 &= \|\mathbf{y}_k - \mathbf{y}^*(\mathbf{x}_{k+1}) - \alpha \nabla_y g(\mathbf{x}_{k+1}, \mathbf{y}_k)\|^2 \\
&= \|\mathbf{y}_k - \mathbf{y}^*(\mathbf{x}_{k+1})\|^2 - 2\alpha \langle \nabla_y g(\mathbf{x}_{k+1}, \mathbf{y}_k), \mathbf{y}_k - \mathbf{y}^*(\mathbf{x}_{k+1}) \rangle \\
&\quad + \alpha^2 \|\nabla_y g(\mathbf{x}_{k+1}, \mathbf{y}_k)\|^2.
\end{aligned} \tag{19}$$

Moreover, from Assumption 2 and following Theorem 2.1.12 in (Nesterov, 2018), we have that

$$\langle \nabla_y g(\mathbf{x}_{k+1}, \mathbf{y}_k), \mathbf{y}_k - \mathbf{y}^*(\mathbf{x}_{k+1}) \rangle \geq \frac{\mu_g L_g}{\mu_g + L_g} \|\mathbf{y}_k - \mathbf{y}^*(\mathbf{x}_{k+1})\|^2 + \frac{1}{\mu_g + L_g} \|\nabla_y g(\mathbf{x}_{k+1}, \mathbf{y}_k)\|^2 \tag{20}$$

The inequality in (19) together with (20) imply that

$$\begin{aligned}
\|\mathbf{y}_{k+1} - \mathbf{y}^*(\mathbf{x}_{k+1})\|^2 &\leq \|\mathbf{y}_k - \mathbf{y}^*(\mathbf{x}_{k+1})\|^2 - \frac{2\alpha \mu_g L_g}{\mu_g + L_g} \|\mathbf{y}_k - \mathbf{y}^*(\mathbf{x}_{k+1})\|^2 \\
&\quad + \left(\alpha^2 - \frac{2\alpha}{\mu_g + L_g}\right) \|\nabla_y g(\mathbf{x}_{k+1}, \mathbf{y}_k)\|^2.
\end{aligned} \tag{21}$$

Setting the step-size $\alpha = \frac{2}{\mu_g + L_g}$ in (21) leads to

$$\|\mathbf{y}_{k+1} - \mathbf{y}^*(\mathbf{x}_{k+1})\|^2 \leq \left(\frac{\mu_g - L_g}{\mu_g + L_g}\right)^2 \|\mathbf{y}_k - \mathbf{y}^*(\mathbf{x}_{k+1})\|^2 \tag{22}$$

Next, recall that $\beta = (L_g - \mu_g)/(L_g + \mu_g)$. Using the triangle inequality and Part (I) of Lemma 1 we conclude that

$$\begin{aligned}
\|\mathbf{y}_{k+1} - \mathbf{y}^*(\mathbf{x}_{k+1})\| &\leq \beta \|\mathbf{y}_k - \mathbf{y}^*(\mathbf{x}_{k+1})\| \\
&\leq \beta \Big[ \|\mathbf{y}_k - \mathbf{y}^*(\mathbf{x}_k)\| + \|\mathbf{y}^*(\mathbf{x}_k) - \mathbf{y}^*(\mathbf{x}_{k+1})\| \Big] \\
&\leq \beta \Big[ \|\mathbf{y}_k - \mathbf{y}^*(\mathbf{x}_k)\| + \mathbf{L_y} \|\mathbf{x}_k - \mathbf{x}_{k+1}\| \Big].
\end{aligned} \tag{23}$$

Moreover, from the update of $\mathbf{x}_{k+1}$ in Algorithm 1 and boundedness of $\mathcal{X}$ we have that $\|\mathbf{x}_{k+1} - \mathbf{x}_k\| \leq \gamma_k D_\mathcal{X}$. Therefore, using this inequality within (23) leads to

$$\|\mathbf{y}_{k+1} - \mathbf{y}^*(\mathbf{x}_{k+1})\| \leq \beta \|\mathbf{y}_k - \mathbf{y}^*(\mathbf{x}_k)\| + \beta \gamma_k \mathbf{L_y} D_\mathcal{X}.$$

Finally, the desired result can be deduced from the above inequality recursively. $\qquad \square$

Previously, in Lemma 4 we quantified how close the approximation $\mathbf{y}_k$ is from the optimal solution $\mathbf{y}^*(\mathbf{x}_k)$ of the inner problem. Now, in the following Lemma, we will find an upper bound for the error of approximating $\mathbf{v}(\mathbf{x}_k)$ via $\mathbf{w}_{k+1}$.

**Lemma 5.** *Let $\{(\mathbf{x}_k, \mathbf{w}_k)\}_{k \geq 0}$ be the sequence generated by Algorithm 1, such that $\gamma_k = \gamma$. Define $\rho_k \triangleq (1 - \eta_k \mu_g)$ and $\mathbf{C}_1 \triangleq L_{yy}^g \frac{C_y^f}{\mu_g} + L_{yy}^f$. Under Assumptions 1 and 2 we have that for any $k \geq 0$,*

$$\|\mathbf{w}_{k+1} - \mathbf{v}(\mathbf{x}_k)\| \leq \rho_k \|\mathbf{w}_k - \mathbf{v}(\mathbf{x}_{k-1})\| + \rho_k \mathbf{C}_\mathbf{v} \gamma D_\mathcal{X} + \eta_k \mathbf{C}_1 \big( \beta^k D_0^y + \mathbf{L}_\mathbf{y} \gamma \frac{\beta}{1-\beta} D_\mathcal{X} \big). \quad (24)$$

*Proof.* From the optimality condition of (8) one can easily verify that $\mathbf{v}(\mathbf{x}_k) = \mathbf{v}(\mathbf{x}_k) - \eta_k \big( \nabla_{yy}^2 g(\mathbf{x}_k, \mathbf{y}^*(\mathbf{x}_k)) \mathbf{v}(\mathbf{x}_k) - \nabla_y f(\mathbf{x}_k, \mathbf{y}^*(\mathbf{x}_k)) \big)$. Now using definition of $\mathbf{w}_{k+1}$ we can write

$$\begin{aligned}
\|\mathbf{w}_{k+1} - \mathbf{v}(\mathbf{x}_k)\| &= \Big\| \Big( \mathbf{w}_k - \eta_k (\nabla_{yy}^2 g(\mathbf{x}_k, \mathbf{y}_k) \mathbf{w}_k - \nabla_y f(\mathbf{x}_k, \mathbf{y}_k)) \Big) - \Big( \mathbf{v}(\mathbf{x}_k) \\
&\quad - \eta_k \big( \nabla_{yy}^2 g(\mathbf{x}_k, \mathbf{y}^*(\mathbf{x}_k)) \mathbf{v}(\mathbf{x}_k) - \nabla_y f(\mathbf{x}_k, \mathbf{y}^*(\mathbf{x}_k)) \big) \Big) \Big\| \\
&= \Big\| \Big( I - \eta_k \nabla_{yy}^2 g(\mathbf{x}_k, \mathbf{y}_k) \Big) (\mathbf{w}_k - \mathbf{v}(\mathbf{x}_k)) - \eta_k \Big( \nabla_{yy}^2 g(\mathbf{x}_k, \mathbf{y}_k) \\
&\quad - \nabla_{yy}^2 g(\mathbf{x}_k, \mathbf{y}^*(\mathbf{x}_k)) \Big) \mathbf{v}(\mathbf{x}_k) + \eta_k \Big( \nabla_y f(\mathbf{x}_k, \mathbf{y}^*(\mathbf{x}_k)) - \nabla_y f(\mathbf{x}_k, \mathbf{y}_k) \Big) \Big\|,
\end{aligned}$$
$$(25)$$

where the last equality is obtained by adding and subtracting the term $(I - \eta_k \nabla_{yy}^2 g(\mathbf{x}_k, \mathbf{y}_k)) \mathbf{v}(\mathbf{x}_k)$. Next, using Assumptions 1 and 2 along with the application of the triangle inequality we obtain

$$\begin{aligned}
\|\mathbf{w}_{k+1} - \mathbf{v}(\mathbf{x}_k)\| &\leq (1 - \eta_k \mu_g) \|\mathbf{w}_k - \mathbf{v}(\mathbf{x}_k)\| + \eta_k L_{yy}^g \|\mathbf{y}_k - \mathbf{y}^*(\mathbf{x}_k)\| \|\mathbf{v}(\mathbf{x}_k)\| \\
&\quad + \eta_k L_{yy}^f \|\mathbf{y}_k - \mathbf{y}^*(\mathbf{x}_k)\|.
\end{aligned}$$
$$(26)$$

Note that $\|\mathbf{v}(\mathbf{x}_k)\| = \|[\nabla_{yy}^2 g(\mathbf{x}, \mathbf{y}^*(\mathbf{x}))]^{-1} \nabla_y f(\mathbf{x}, \mathbf{y}^*(\mathbf{x}))\| \leq \frac{C_y^f}{\mu_g}$. Now, by adding and subtracting $\mathbf{v}(\mathbf{x}_{k-1})$ to the term $\|\mathbf{w}_k - \mathbf{v}(\mathbf{x}_k)\|$ followed by triangle inequality application we can conclude that

$$\begin{aligned}
\|\mathbf{w}_{k+1} - \mathbf{v}(\mathbf{x}_k)\| &\leq (1 - \eta_k \mu_g) \|\mathbf{w}_k - \mathbf{v}(\mathbf{x}_{k-1})\| + (1 - \eta_k \mu_g) \|\mathbf{v}(\mathbf{x}_{k-1}) - \mathbf{v}(\mathbf{x}_k)\| \\
&\quad + \eta_k \Big( L_{yy}^g \frac{C_y^f}{\mu_g} + L_{yy}^f \Big) \|\mathbf{y}_k - \mathbf{y}^*(\mathbf{x}_k)\|.
\end{aligned}$$
$$(27)$$

Therefore, using the result of Lemma 4, we can further bound inequality (27) as follows

$$\begin{aligned}
\|\mathbf{w}_{k+1} - \mathbf{v}(\mathbf{x}_k)\| &\leq (1 - \eta_k \mu_g) \|\mathbf{w}_k - \mathbf{v}(\mathbf{x}_{k-1})\| + (1 - \eta_k \mu_g) \mathbf{C}_\mathbf{v} \|\mathbf{x}_{k-1} - \mathbf{x}_k\| \\
&\quad + \eta_k \mathbf{C}_1 \|\mathbf{y}_k - \mathbf{y}^*(\mathbf{x}_k)\| \\
&\leq \rho_k \|\mathbf{w}_k - \mathbf{v}(\mathbf{x}_{k-1})\| + \rho_k \mathbf{C}_\mathbf{v} \gamma D_\mathcal{X} + \eta_k \mathbf{C}_1 \big( \beta^k D_0^y + \mathbf{L}_\mathbf{y} \gamma \frac{\beta}{1-\beta} D_\mathcal{X} \big)
\end{aligned}$$
$$(28)$$

where the last inequality follows from the boundedness assumption of set $\mathcal{X}$, recalling that $D_0^y = \|\mathbf{y}_0 - \mathbf{y}^*(\mathbf{x}_0)\|$, and the fact that $\sum_{i=0}^{k-1} \beta^{k-i} \leq \frac{\beta}{1-\beta}$. $\qquad \square$

**Lemma 6.** *Let $\{(\mathbf{x}_k, \mathbf{w}_k)\}_{k \geq 0}$ be the sequence generated by Algorithm 1 with step-size $\eta_k = \eta < \frac{1-\beta}{\mu_g}$ where $\beta$ is defined in Lemma 4. Suppose that Assumption 2 holds and $\mathbf{v}(\mathbf{x}_{-1}) = \mathbf{v}(\mathbf{x}_0)$, then for any $K \geq 1$,*

$$\|\mathbf{w}_K - \mathbf{v}(\mathbf{x}_{K-1})\| \leq \rho^K \|\mathbf{w}_0 - \mathbf{v}(\mathbf{x}_0)\| + \frac{\gamma \rho \mathbf{C}_\mathbf{v} D_\mathcal{X}}{1-\rho} + \frac{\eta \mathbf{C}_1 D_0^y \rho^{K+1}}{\rho - \beta} + \frac{\gamma \eta \beta \mathbf{C}_1 \mathbf{L}_\mathbf{y} D_\mathcal{X}}{(1-\rho)(1-\beta)}, \quad (29)$$

*where $\rho \triangleq 1 - \eta \mu_g$.*

*Proof.* Applying the result of Lemma 5 recursively for $k = 0$ to $K - 1$, one can conclude that

$$\begin{aligned}
\|\mathbf{w}_K - \mathbf{v}(\mathbf{x}_{K-1})\| &\leq \rho^K \|\mathbf{w}_0 - \mathbf{v}(\mathbf{x}_0)\| + \mathbf{C}_\mathbf{v} \gamma D_\mathcal{X} \sum_{i=1}^K \rho^i + \eta \mathbf{C}_1 \sum_{i=0}^K \big( \beta^i D_0^y + \gamma \mathbf{L}_\mathbf{y} D_\mathcal{X} \frac{\beta}{1-\beta} \big) \rho^{K-i} \\
&\leq \rho^K \|\mathbf{w}_0 - \mathbf{v}(\mathbf{x}_0)\| + \frac{\rho}{1-\rho} \mathbf{C}_\mathbf{v} \gamma D_\mathcal{X} + \eta \mathbf{C}_1 D_0^y \big( \sum_{i=0}^K \beta^i \rho^{K-i} \big) \\
&\quad + \frac{\gamma \eta \beta \mathbf{C}_1 \mathbf{L}_\mathbf{y} D_\mathcal{X}}{1-\beta} \sum_{i=0}^K \rho^{K-i},
\end{aligned}$$
$$(30)$$

where the last inequality is obtained by noting that $\sum_{i=1}^{K} \rho^i \leq \frac{\rho}{1-\rho}$. Finally, the choice $\eta < \frac{1-\beta}{\mu_g}$ implies that $\beta < \rho$, hence, $\sum_{i=0}^{K} (\frac{\beta}{\rho})^i \leq \frac{\rho}{\rho-\beta}$ which leads to the desired result. $\qquad \square$

## B.1 PROOF OF LEMMA 2

We begin the proof by considering the definition of $\nabla \ell(\mathbf{x}_k)$ and $F_k$ followed by a triangle inequality to obtain

$$
\|\nabla \ell(\mathbf{x}_k) - F_k\| \leq \|\nabla_x f(\mathbf{x}_k, \mathbf{y}^*(\mathbf{x}_k)) - \nabla_x f(\mathbf{x}_k, \mathbf{y}_k)\| \\
+ \|\nabla_{yx}^2 g(\mathbf{x}_k, \mathbf{y}_k) \mathbf{w}_{k+1} - \nabla_{yx}^2 g(\mathbf{x}_k, \mathbf{y}^*(\mathbf{x}_k)) \mathbf{v}(\mathbf{x}_k)\| \tag{31}
$$

Combining Assumption 1-(i) together with adding and subtracting $\nabla_{yx}^2 g(\mathbf{x}_k, \mathbf{y}_k) \mathbf{v}(\mathbf{x}_k)$ to the second term of RHS lead to

$$
\|\nabla \ell(\mathbf{x}_k) - F_k\| \leq L_{xy}^f \|\mathbf{y}_k - \mathbf{y}^*(\mathbf{x}_k)\| + \|\nabla_{yx}^2 g(\mathbf{x}_k, \mathbf{y}_k)(\mathbf{w}_{k+1} - \mathbf{v}(\mathbf{x}_k)) + (\nabla_{yx}^2 g(\mathbf{x}_k, \mathbf{y}_k) \\
- \nabla_{yx}^2 g(\mathbf{x}_k, \mathbf{y}^*(\mathbf{x}_k))) \mathbf{v}(\mathbf{x}_k)\|
$$

$$
\leq L_{xy}^f \|\mathbf{y}_k - \mathbf{y}^*(\mathbf{x}_k)\| + C_{yx}^g \|\mathbf{w}_{k+1} - \mathbf{v}(\mathbf{x}_k)\| + L_{yx}^g \frac{C_y^f}{\mu_g} \|\mathbf{y}_k - \mathbf{y}^*(\mathbf{x}_k)\| \tag{32}
$$

where the last inequality is obtained using Assumption 2 and the triangle inequality. Next, utilizing Lemma 4 and 6 we can further provide upper-bounds for the term in RHS of (32) as follows

$$
\|\nabla \ell(\mathbf{x}_k) - F_k\| \leq \mathbf{C}_2 \left( \beta^k D_0^y + \frac{\gamma \beta \mathbf{L_y} D_{\mathcal{X}}}{1-\beta} \right) + C_{yx}^g \left( \rho^{k+1} \|\mathbf{w}_0 - \mathbf{v}(\mathbf{x}_0)\| + \frac{\gamma \rho \mathbf{C_v} D_{\mathcal{X}}}{1-\rho} \right. \\
\left. + \frac{\eta \mathbf{C}_1 D_0^y \rho^{k+2}}{\rho - \beta} + \frac{\gamma \eta \beta \mathbf{C}_1 \mathbf{L_y} D_{\mathcal{X}}}{(1-\rho)(1-\beta)} \right).
$$

$\qquad \square$

## B.2 IMPROVEMENT IN ONE STEP

In the following, we characterize the improvement of the objective function $\ell(\mathbf{x})$ after taking one step of Algorithm 1.

**Lemma 7.** *Let $\{\mathbf{x}_k\}_{k=0}^{K}$ be the sequence generated by Algorithm 1. Suppose Assumptions 1 and 2 hold and $\gamma_k = \gamma$, then for any $k \geq 0$ we have*

$$
\ell(\mathbf{x}_{k+1}) \leq \ell(\mathbf{x}_k) - \gamma \mathcal{G}(\mathbf{x}_k) + \gamma \mathbf{C}_2 \beta^k D_0^y D_{\mathcal{X}} + \frac{\gamma^2 \mathbf{C}_2 D_{\mathcal{X}}^2 \mathbf{L_y} \beta}{1-\beta} + C_{yx}^g \left[ \gamma D_{\mathcal{X}} \rho^{k+1} \|\mathbf{w}_0 - \mathbf{v}(\mathbf{x}_0)\| \right.
$$

$$
\left. + \frac{\gamma^2 D_{\mathcal{X}}^2 \rho \mathbf{C_v}}{1-\rho} + \frac{\gamma D_{\mathcal{X}} D_0^y \mathbf{C}_1 \eta \rho^{k+2}}{\rho - \beta} + \frac{\gamma^2 D_{\mathcal{X}}^2 \mathbf{L_y} \mathbf{C}_1 \beta \eta}{(1-\beta)(1-\rho)} \right] + \frac{1}{2} \mathbf{L}_\ell \gamma^2 D_{\mathcal{X}}^2. \tag{33}
$$

*Proof.* Note that according to Lemma 1-(III), $\ell(\cdot)$ has a Lipschitz continuous gradient which implies that

$$
\ell(\mathbf{x}_{k+1}) \leq \ell(\mathbf{x}_k) + \gamma \langle \nabla \ell(\mathbf{x}_k), \mathbf{s}_k - \mathbf{x}_k \rangle + \frac{1}{2} \mathbf{L}_\ell \gamma^2 \|\mathbf{s}_k - \mathbf{x}_k\|^2
$$

$$
= \ell(\mathbf{x}_k) + \gamma \langle F_k, \mathbf{s}_k - \mathbf{x}_k \rangle + \gamma \langle \nabla \ell(\mathbf{x}_k) - F_k, \mathbf{s}_k - \mathbf{x}_k \rangle + \frac{1}{2} \mathbf{L}_\ell \gamma^2 \|\mathbf{s}_k - \mathbf{x}_k\|^2, \tag{34}
$$

where the last inequality follows from adding and subtracting the term $\gamma \langle F_k, \mathbf{s}_k - \mathbf{x}_k \rangle$ to the RHS. Define $\mathbf{s}_k' = \arg\max_{\mathbf{s} \in \mathcal{X}} \{ \langle \nabla \ell(\mathbf{x}_k), \mathbf{x}_k - \mathbf{s} \rangle \}$ and observe that $\mathcal{G}(\mathbf{x}_k) = \langle \nabla \ell(\mathbf{x}_k), \mathbf{x}_k - \mathbf{s}_k' \rangle$ by Definition 1. Using the definition of $\mathbf{s}_k$, we can immediately observe that

$$
\langle F_k, \mathbf{s}_k - \mathbf{x}_k \rangle = \min_{\mathbf{s} \in \mathcal{X}} \langle F_k, \mathbf{s} - \mathbf{x}_k \rangle
$$

$$
\leq \langle F_k, \mathbf{s}_k' - \mathbf{x}_k \rangle
$$

$$
= \langle \nabla \ell(\mathbf{x}_k), \mathbf{s}_k' - \mathbf{x}_k \rangle + \langle F_k - \nabla \ell(\mathbf{x}_k), \mathbf{s}_k' - \mathbf{x}_k \rangle
$$

$$
= -\mathcal{G}(\mathbf{x}_k) + \langle F_k - \nabla \ell(\mathbf{x}_k), \mathbf{s}_k' - \mathbf{x}_k \rangle. \tag{35}
$$

Next, combining (34) with (35) followed by the Cauchy-Schwartz inequality leads to

$$\ell(\mathbf{x}_{k+1}) \le \ell(\mathbf{x}_k) - \gamma\mathcal{G}(\mathbf{x}_k) + \gamma\|\nabla\ell(\mathbf{x}_k) - F_k\|\|\mathbf{s}_k - \mathbf{s}_k'\| + \frac{1}{2}\mathbf{L}_\ell\gamma^2\|\mathbf{s}_k - \mathbf{x}_k\|^2. \tag{36}$$

Finally, using the result of the Lemma 2 together with the boundedness assumption of set $\mathcal{X}$ we conclude the desired result. $\qquad\square$

## C  PROOF OF THEOREM 1

Since $\ell$ is convex, from the definition of $\mathcal{G}(\mathbf{x}_k)$ in (4) we have

$$\mathcal{G}(\mathbf{x}_k) = \max_{\mathbf{s}\in\mathcal{X}}\{\langle\nabla\ell(\mathbf{x}_k), \mathbf{x}_k - \mathbf{s}\rangle\} \ge \langle\nabla\ell(\mathbf{x}_k), \mathbf{x}_k - \mathbf{x}^*\rangle \ge \ell(\mathbf{x}_k) - \ell(\mathbf{x}^*). \tag{37}$$

We assume a fixed step-size in Theorem 1 and we set $\gamma_k = \gamma$. Combining the result of Lemma 7 with (37) leads to

$$\ell(\mathbf{x}_{k+1}) \le \ell(\mathbf{x}_k) - \gamma(\ell(\mathbf{x}_k) - \ell(\mathbf{x}^*)) + \gamma\mathbf{C}_2\beta^k D_0^y D_\mathcal{X} + \frac{\gamma^2\mathbf{C}_2 D_\mathcal{X}^2\mathbf{L}_\mathbf{y}\beta}{1-\beta} + C_{yx}^g\Big[\gamma D_\mathcal{X}\rho^{k+1}\|\mathbf{w}_0 - \mathbf{v}(\mathbf{x}_0)\|$$
$$+ \frac{\gamma^2 D_\mathcal{X}^2\rho\mathbf{C}_\mathbf{v}}{1-\rho} + \frac{\gamma D_\mathcal{X} D_0^y\mathbf{C}_1\eta\rho^{k+2}}{\rho-\beta} + \frac{\gamma^2 D_\mathcal{X}^2\mathbf{L}_\mathbf{y}\mathbf{C}_1\beta\eta}{(1-\beta)(1-\rho)}\Big] + \frac{1}{2}\mathbf{L}_\ell\gamma^2 D_\mathcal{X}^2. \tag{38}$$

Subtracting $\ell(\mathbf{x}^*)$ from both sides, we get

$$\ell(\mathbf{x}_{k+1}) - \ell(\mathbf{x}^*) \le (1-\gamma)(\ell(\mathbf{x}_k) - \ell(\mathbf{x}^*)) + \mathcal{R}_k(\gamma), \tag{39}$$

where

$$\mathcal{R}_k(\gamma) \triangleq \gamma\mathbf{C}_2\beta^k D_0^y D_\mathcal{X} + \frac{\gamma^2\mathbf{C}_2 D_\mathcal{X}^2\mathbf{L}_\mathbf{y}\beta}{1-\beta} + C_{yx}^g\Big[\gamma D_\mathcal{X}\rho^{k+1}\|\mathbf{w}_0 - \mathbf{v}(\mathbf{x}_0)\|$$
$$+ \frac{\gamma^2 D_\mathcal{X}^2\rho\mathbf{C}_\mathbf{v}}{1-\rho} + \frac{\gamma D_\mathcal{X} D_0^y\mathbf{C}_1\eta\rho^{k+2}}{\rho-\beta} + \frac{\gamma^2 D_\mathcal{X}^2\mathbf{L}_\mathbf{y}\mathbf{C}_1\beta\eta}{(1-\beta)(1-\rho)}\Big] + \frac{1}{2}\mathbf{L}_\ell\gamma^2 D_\mathcal{X}^2. \tag{40}$$

Continuing (39) recursively leads to the desired result. $\qquad\square$

## D  PROOF OF COROLLARY 1

We start the proof by using the result of the Theorem 1, i.e.,

$$\ell(\mathbf{x}_K) - \ell(\mathbf{x}^*) \le (1-\gamma)^K(\ell(\mathbf{x}_0) - \ell(\mathbf{x}^*)) + \sum_{k=0}^{K-1}(1-\gamma)^{K-k}\mathcal{R}_k(\gamma). \tag{41}$$

Note that

$$\sum_{k=0}^{K-1}(1-\gamma)^{K-k}\mathcal{R}_k(\gamma)$$

$$= \mathbf{C}_2 D_0^y D_\mathcal{X}\Big[\sum_{k=0}^{K-1}(1-\gamma)^{K-k}\gamma\beta^k\Big] + \frac{\mathbf{C}_2 D_\mathcal{X}^2\mathbf{L}_\mathbf{y}\beta}{1-\beta}\Big[\sum_{k=0}^{K-1}(1-\gamma)^{K-k}\gamma^2\Big]$$

$$+ C_{yx}^g\Big(\rho D_\mathcal{X}\|\mathbf{w}_0 - \mathbf{v}(\mathbf{x}_0)\|\Big[\sum_{k=0}^{K-1}(1-\gamma)^{K-k}\gamma\rho^k\Big] + \frac{D_\mathcal{X}^2\rho\mathbf{C}_\mathbf{v}}{1-\rho}\Big[\sum_{k=0}^{K-1}(1-\gamma)^{K-k}\gamma^2\Big]$$

$$+ \frac{D_\mathcal{X} D_0^y\mathbf{C}_1\eta\rho^2}{\rho-\beta}\Big[\sum_{k=0}^{K-1}(1-\gamma)^{K-k}\gamma\rho^k\Big] + \frac{D_\mathcal{X}^2\mathbf{L}_\mathbf{y}\mathbf{C}_1\beta\eta}{(1-\beta)(1-\rho)}\Big[\sum_{k=0}^{K-1}(1-\gamma)^{K-k}\gamma^2\Big]\Big)$$

$$+ \frac{1}{2}\mathbf{L}_\ell D_\mathcal{X}^2\Big[\sum_{k=0}^{K-1}(1-\gamma)^{K-k}\gamma^2\Big].$$

Moreover, one can easily verify that $\sum_{k=0}^{K-1}(1-\gamma)^{K-k}\gamma^2 \le \gamma(1-\gamma)$ and $\sum_{k=0}^{K-1}(1-\gamma)^{K-k}\gamma\rho^k \le \frac{\gamma(1-\gamma)}{|1-\gamma-\rho|}$ from which together with the above inequality we conclude that

$$
\sum_{k=0}^{K-1}(1-\gamma)^{K-k}\mathcal{R}_k(\gamma)
$$

$$
\le \frac{\mathbf{C}_2 D_0^y D_{\mathcal{X}}\gamma(1-\gamma)}{|1-\gamma-\beta|} + \frac{\mathbf{C}_2 D_{\mathcal{X}}^2 \mathbf{L_y}\beta\gamma(1-\gamma)}{1-\beta} + C_{yx}^g\Big(\frac{D_{\mathcal{X}}\rho\gamma(1-\gamma)}{|1-\gamma-\rho|}\|\mathbf{w}_0 - \mathbf{v}(\mathbf{x}_0)\|
$$

$$
+ \frac{D_{\mathcal{X}}^2 \mathbf{C_v}\rho\gamma(1-\gamma)}{1-\rho} + \frac{D_{\mathcal{X}} D_0^y \mathbf{C}_1\eta\rho^2\gamma(1-\gamma)}{(\rho-\beta)|1-\gamma-\rho|} + \frac{D_{\mathcal{X}}^2 \mathbf{L_y}\mathbf{C}_1\eta\beta\gamma(1-\gamma)}{(1-\beta)(1-\rho)}\Big) + \frac{1}{2}\mathbf{L}_\ell D_{\mathcal{X}}^2\gamma(1-\gamma)
$$

$$
= \mathcal{O}\Big(\frac{\mathbf{C_v}\rho}{1-\rho}\gamma + \frac{\mathbf{L_y}\mathbf{C}_1\beta}{(1-\beta)(1-\rho)}\gamma\Big). \tag{42}
$$

Using the above inequality within (41) we conclude that $\ell(\mathbf{x}_K) - \ell(\mathbf{x}^*) \le (1-\gamma)^K(\ell(\mathbf{x}_0) - \ell(\mathbf{x}^*)) + \mathcal{O}(\frac{\mathbf{C_v}\rho}{1-\rho}\gamma + \frac{\mathbf{L_y}\mathbf{C}_1\beta}{(1-\beta)(1-\rho)}\gamma)$ where $\mathbf{C_v} = \mathcal{O}(\kappa_g^3)$, $\mathbf{C}_1 = \mathcal{O}(\kappa_g^2)$, $\mathbf{L_y} = \mathcal{O}(\kappa_g)$ as shown in Lemma 3 and $\min\{1-\rho, 1-\beta\} = \Omega(\frac{1}{\kappa_g})$ as shown in Lemma 2. Next, we show that by selecting $\gamma = \log(K)/K$ we have that $(1-\gamma)^K \le 1/K$. In fact, for any $x > 0$, $\log(x) \ge 1 - \frac{1}{x}$ which implies that $\log(\frac{1}{1-\gamma}) \ge \gamma = \log(K)/K$, hence, $(\frac{1}{1-\gamma})^K \ge K$. Putting the pieces together we conclude that $\ell(\mathbf{x}_K) - \ell(\mathbf{x}^*) = \mathcal{O}((1-\gamma)^K(\ell(\mathbf{x}_0) - \ell(\mathbf{x}^*)) + \gamma\kappa_g^5) = \tilde{\mathcal{O}}(\kappa_g^5/K)$, which leads to an iteration complexity of $\tilde{\mathcal{O}}(\kappa_g^5\epsilon^{-1})$.

Furthermore, assuming that $\nabla_y f(\mathbf{x}, \cdot)$ is uniformly bounded for any $\mathbf{x} \in \mathcal{X}$, we conclude that $C_y^f = \mathcal{O}(1)$, hence, $\mathbf{C}_1 = \mathcal{O}(\kappa_g)$ from which we have that $\ell(\mathbf{x}_K) - \ell(\mathbf{x}^*) = \mathcal{O}((1-\gamma)^K(\ell(\mathbf{x}_0) - \ell(\mathbf{x}^*)) + \gamma\kappa_g^4)$. Therefore, selecting $\gamma = \log(K)/K$ implies that $\ell(\mathbf{x}_K) - \ell(\mathbf{x}^*) = \mathcal{O}(\kappa_g^4/K)$ which leads to an iteration complexity of $\mathcal{O}(\kappa_g^4\epsilon^{-1})$. $\qquad\square$

## E  PROOF OF THEOREM 2

Recall that from Lemma 7 we have

$$
\mathcal{G}(\mathbf{x}_k) \le \frac{\ell(\mathbf{x}_k) - \ell(\mathbf{x}_{k+1})}{\gamma} + \mathbf{C}_2\beta^k D_0^y D_{\mathcal{X}} + \frac{\gamma\mathbf{C}_2 D_{\mathcal{X}}^2 \mathbf{L_y}\beta}{1-\beta} + C_{yx}^g\Big[D_{\mathcal{X}}\rho^{k+1}\|\mathbf{w}_0 - \mathbf{v}(\mathbf{x}_0)\|
$$

$$
+ \frac{\gamma D_{\mathcal{X}}^2\rho\mathbf{C_v}}{1-\rho} + \frac{D_{\mathcal{X}} D_0^y \mathbf{C}_1\eta\rho^{k+2}}{\rho-\beta} + \frac{\gamma D_{\mathcal{X}}^2 \mathbf{L_y}\mathbf{C}_1\beta\eta}{(1-\beta)(1-\rho)}\Big] + \frac{1}{2}\mathbf{L}_\ell\gamma D_{\mathcal{X}}^2.
$$

Summing both sides of the above inequality from $k = 0$ to $K-1$, we get

$$
\sum_{k=0}^{K-1}\mathcal{G}(\mathbf{x}_k) \le \frac{\ell(\mathbf{x}_0) - \ell(\mathbf{x}_K)}{\gamma} + \frac{\mathbf{C}_2 D_0^y D_{\mathcal{X}}}{1-\beta} + K\frac{\gamma\mathbf{C}_2 D_{\mathcal{X}}^2 \mathbf{L_y}\beta}{1-\beta} + C_{yx}^g\Big[\frac{\rho D_{\mathcal{X}}\|\mathbf{w}_0 - \mathbf{v}(\mathbf{x}_0)\|}{1-\rho}
$$

$$
+ K\frac{\gamma D_{\mathcal{X}}^2\rho\mathbf{C_v}}{1-\rho} + \frac{D_{\mathcal{X}} D_0^y \mathbf{C}_1\eta\rho^2}{(1-\rho)(\rho-\beta)} + K\frac{\gamma D_{\mathcal{X}}^2 \mathbf{L_y}\mathbf{C}_1\beta\eta}{(1-\beta)(1-\rho)}\Big] + \frac{K}{2}\mathbf{L}_\ell\gamma D_{\mathcal{X}}^2,
$$

where in the above inequality we use the fact that $\sum_{i=0}^K \beta^i \le \frac{1}{1-\beta}$. Next, dividing both sides of the above inequality by $K$ and denoting the smallest gap function over the iterations from $k = 0$ to $K-1$, i.e.,

$$
\mathcal{G}_{k^*} \triangleq \min_{0 \le k \le K-1} \mathcal{G}(\mathbf{x}_k) \le \frac{1}{K}\sum_{k=0}^{K-1}\mathcal{G}(\mathbf{x}_k),
$$

imply that

$$
\mathcal{G}_{k^*} \le \frac{\ell(\mathbf{x}_0) - \ell(\mathbf{x}_K)}{K\gamma} + \frac{\gamma\mathbf{C}_2 D_{\mathcal{X}} \mathbf{L_y}\beta}{1-\beta} + \frac{\gamma D_{\mathcal{X}}^2\rho\mathbf{C_v}C_{yx}^g\rho}{1-\rho} + \frac{\gamma D_{\mathcal{X}}^2 C_{yx}^g \mathbf{L_y}\mathbf{C}_1\beta\eta}{(1-\beta)(1-\rho)} + \frac{1}{2}\mathbf{L}_\ell\gamma D_{\mathcal{X}}^2
$$

$$
+ \frac{\mathbf{C}_2 D_0^y D_{\mathcal{X}}\beta}{K(1-\beta)} + \frac{D_{\mathcal{X}} C_{yx}^g\rho\|\mathbf{w}_0 - \mathbf{v}(\mathbf{x}_0)\|}{K(1-\rho)} + \frac{D_{\mathcal{X}} D_0^y C_{yx}^g \mathbf{C}_1\eta\rho^2}{K(1-\beta)(1-\rho)}. \tag{43}
$$

$$\square$$

## F  PROOF OF COROLLARY 2

We begin the proof by using the result of the Theorem 2.

$$
\begin{aligned}
\mathcal{G}_{k^*} &\leq \frac{\ell(\mathbf{x}_0) - \ell(\mathbf{x}_K)}{K\gamma} + \frac{\gamma \mathbf{C}_2 D_\mathcal{X} \mathbf{L_y}\beta}{1-\beta} + \frac{\gamma D_\mathcal{X}^2 \rho \mathbf{C_v} C_{yx}^g \rho}{1-\rho} + \frac{\gamma D_\mathcal{X}^2 C_{yx}^g \mathbf{L_y} \mathbf{C}_1 \beta\eta}{(1-\beta)(1-\rho)} + \frac{1}{2}\mathbf{L}_\ell \gamma D_\mathcal{X}^2 \\
&\quad + \frac{\mathbf{C}_2 D_0^y D_\mathcal{X}\beta}{K(1-\beta)} + \frac{D_\mathcal{X} C_{yx}^g \rho \|\mathbf{w}_0 - \mathbf{v}(\mathbf{x}_0)\|}{K(1-\rho)} + \frac{D_\mathcal{X} D_0^y C_{yx}^g \mathbf{C}_1 \eta\rho^2}{K(1-\beta)(1-\rho)} \\
&= \mathcal{O}\left( \frac{1}{K\gamma} + \frac{\gamma \mathbf{C}_2 \mathbf{L_y}\beta}{1-\beta} + \frac{\gamma \mathbf{L_y} \mathbf{C}_1 \beta}{(1-\beta)(1-\rho)} \right)
\end{aligned}
$$

The desired result follows immediately from (43) and the fact that $\ell(\mathbf{x}^*) \leq \ell(\mathbf{x}_K)$. Moreover, similar to the proof of Corollary 1 we have that $\mathbf{C_v} = \mathcal{O}(\kappa_g^3)$, $\mathbf{C}_1 = \mathcal{O}(\kappa_g^2)$, $\mathbf{L_y} = \mathcal{O}(\kappa_g)$, and $\min\{1-\rho, 1-\beta\} = \Omega(\frac{1}{\kappa_g})$. Hence, by choosing $\gamma = 1/(\kappa_g^{2.5}\sqrt{K})$, we obtain that $\mathcal{G}_k^* = \mathcal{O}(\frac{1}{K\gamma} + \gamma\kappa_g^5) = \mathcal{O}(\kappa_g^{2.5}/\sqrt{K})$, which leads to an iteration complexity of $\mathcal{O}(\kappa_g^5\epsilon^{-2})$.

Furthermore, assuming that $\nabla_y f(x, y)$ is uniformly bounded, we conclude that $C_y^f = \mathcal{O}(1)$, hence, $\mathbf{C}_1 = \mathcal{O}(\kappa_g)$ from which we have that $\mathcal{G}_{k^*} = \mathcal{O}(\frac{1}{K\gamma} + \gamma\kappa_g^4)$. Therefore, selecting $\gamma = 1/(\kappa_g^2\sqrt{K})$ implies that $\mathcal{G}_{k^*} = \mathcal{O}(\kappa_g^2/\sqrt{K})$ which leads to an iteration complexity of $\mathcal{O}(\kappa_g^4\epsilon^{-2})$.  $\square$

## G  ADDITIONAL EXPERIMENTS

In this section, we provide more details about the experiments conducted in section 5 as well as some additional experiments.

### G.1  EXPERIMENT DETAILS

In this section, we include more details of the numerical experiments in Section 5. The MATLAB code is also included in the supplementary material.

For completeness, we briefly review the update rules of SBFW (Akhtar et al., 2022) and TTSA (Hong et al., 2020) for the setting considered in problem (1). In the following, we use $\mathcal{P}_\mathcal{X}(\cdot)$ to denote the Euclidean projection onto the set $\mathcal{X}$.

Each iteration of SBFW has the following updates:

$$
\begin{aligned}
\mathbf{y}_k &= \mathbf{y}_{k-1} - \delta_k \nabla_y g(\mathbf{x}_{k-1}, \mathbf{y}_{k-1}), \\
\mathbf{d}_k &= (1-\rho_k)(\mathbf{d}_{k-1} - h(\mathbf{x}_{k-1}, \mathbf{y}_{k-1})) + h(\mathbf{x}_k, \mathbf{y}_k), \\
\mathbf{s}_k &= \operatorname*{argmin}_{\mathbf{s} \in \mathcal{X}} \langle \mathbf{s}, \mathbf{d}_k \rangle, \\
\mathbf{x}_{k+1} &= (1-\eta_k)\mathbf{x}_k + \eta_k \mathbf{s}_k
\end{aligned}
$$

Based on the theoretical analysis in (Akhtar et al., 2022), $\rho_k = \frac{2}{k^{1/2}}$, $\eta_k = \frac{2}{(k+1)^{3/4}}$, and $\delta_k = \frac{a_0}{k^{1/2}}$ where $a_0 = \min\left\{\frac{2}{3\mu_g}, \frac{\mu_g}{2L_g^2}\right\}$. Moreover, $h(\mathbf{x}_k, \mathbf{y}_k)$ is a biased estimator of the surrogate $\ell(\mathbf{x}_k)$ which can be computed as follows

$$
h(\mathbf{x}_k, \mathbf{y}_k) = \nabla_x f(\mathbf{x}_k, \mathbf{y}_k) - M(\mathbf{x}_k, \mathbf{y}_k)\nabla_y f(\mathbf{x}_k, \mathbf{y}_k),
$$

where the term $M(\mathbf{x}_k, \mathbf{y}_k)$ is a biased estimation of $[\nabla_{yy}^2 g(\mathbf{x}_k, \mathbf{y}_k)]^{-1}$ with bounded variance whose explicit form is

$$
M(\mathbf{x}_k, \mathbf{y}_k) = \nabla_{yx}^2 g(\mathbf{x}_k, \mathbf{y}_k) \times \left[ \frac{k}{L_g} \Pi_{i=1}^l \left( I - \frac{1}{L_g} \nabla_{yy}^2 g(x_k, y_k) \right) \right],
$$

and $l \in \{1, \ldots, k\}$ is an integer selected uniformly at random.

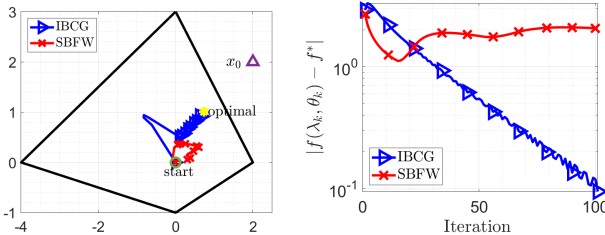

Figure 3: The performance of IBCG (blue) vs SBFW (red) on Problem (44) when $\mu_g = 1$. Plots from left to right are trajectories of $\theta_k$ and $f(\lambda_k, \theta_k) - f^*$.

The steps of TTSA algorithm are given by

$$\mathbf{y}_{k+1} = \mathbf{y}_k - \beta h_k^g,$$
$$\mathbf{x}_{k+1} = \mathcal{P}_{\mathcal{X}}(\mathbf{x}_k - \alpha h_k^f),$$
$$h_k^g = \nabla_y g(\mathbf{x}_k, \mathbf{y}_k),$$
$$h_k^f = \nabla_x f(\mathbf{x}_k, \mathbf{y}_k) - \nabla_{yx}^2 g(\mathbf{x}_k, \mathbf{y}_k) \times \left[ \frac{t_{max}(k)c_h}{L_g} \Pi_{i=1}^p \left( I - \frac{c_h}{L_g} \nabla_{yy}^2 g(\mathbf{x}_k, \mathbf{y}_k) \right) \right] \nabla_y f(\mathbf{x}_k, \mathbf{y}_k),$$

where based on the theory we define $L = L_x^f + \frac{L_y^f C_{yx}^g}{\mu_g} + C_y^f \left( \frac{L_{yx}^g}{\mu_g} + \frac{L_{yy}^g C_{yx}^g}{\mu_g^2} \right)$, and $L_y = \frac{C_{yx}^g}{\mu_g}$, then set $\alpha = \min \left\{ \frac{\mu_g^2}{8L_y L L_g^2}, \frac{1}{4 L_y L} K^{-3/5} \right\}$, $\beta = \min \left\{ \frac{\mu_g}{L_g^2}, \frac{2}{\mu_g} K^{-2/5} \right\}$, $t_{max}(k) = \frac{L_g}{\mu_g} \log(k+1)$, $p \in \{0, \dots, t_{max}(k) - 1\}$, and $c_h \in (0, 1]$.

### G.2 Toy example

Here we consider a variation of coreset problem in a two-dimensional space to illustrate the numerical stability of our proposed method. Given a point $x_0 \in \mathbb{R}^2$, the goal is to find the closest point to $x_0$ such that under a linear map it lies within the convex hull of given points $\{x_1, x_2, x_3, x_4\} \subset \mathbb{R}^2$. Let $A \in \mathbb{R}^{2 \times 2}$ represents the linear map, $X \triangleq [x_1, x_2, x_3, x_4] \in \mathbb{R}^{2 \times 4}$, and $\Delta_4 \triangleq \{\lambda \in \mathbb{R}^4 | \langle \lambda, 1 \rangle = 1, \lambda \geq 0\}$ be the standard simplex set. This problem can be formulated as the following bilevel optimization problem

$$\min_{\lambda \in \Delta_4} \frac{1}{2} \|\theta(\lambda) - x_0\|^2 \quad \text{s.t.} \quad \theta(\lambda) \in \operatorname*{argmin}_{\theta \in \mathbb{R}^2} \frac{1}{2} \|A\theta - X\lambda\|^2. \tag{44}$$

We set the target $x_0 = (2, 2)$ and choose starting points as $\theta_0 = (0, 0)$ and $\lambda_0 = \mathbf{1}_4/4$. We implemented our proposed method and compared it with SBFW (Akhtar et al., 2022). It should be noted that in the SBFW method, they used a biased estimation for $[\nabla_{yy}^2 g(\lambda, \theta)]^{-1} = (A^\top A)^{-1}$ whose bias is upper bounded by $\frac{2}{\mu_g}$ (see (Ghadimi & Wang, 2018, Lemma 3.2)). Figure 3 illustrates the iteration trajectories of both methods for $\mu_g = 1$ and $K = 10^2$. The step-sizes for both methods are selected as suggested by their theoretical analysis. We observe that our method converges to the optimal solution while SBFW fails to converge. This situation for SBFW exacerbates for smaller values of $\mu_g$.

Figure 4 illustrates the iteration trajectories of both methods for $\mu_g = 0.1$ and $K = 10^3$ in which we also included SBFW method whose Hessian inverse matrix is explicitly provided in the algorithm. The step-sizes for both methods are selected as suggested by their theoretical analysis. Despite incorporating the Hessian inverse matrix in the SBFW method, the algorithm's effectiveness is compromised by excessively conservative step-sizes, as dictated by the theoretical result. Consequently, the algorithm fails to converge to the optimal point effectively. Regarding this issue, we tune their step-sizes, i.e., scale the parameter $\delta$ and $\eta$ in their method by a factor of 5 and 0.1, respectively. By tuning the parameters we can see in Figure 5 that the SBFW with Hessian inverse matrix algorithm has a better performance and converges to the optimal solution. In fact, using the Hessian inverse as well as tuning the step-sizes their method converges to the optimal solution while our method always shows a consistent and robust behavior.

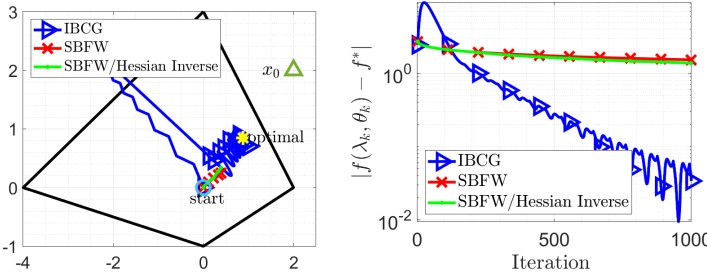

Figure 4: The performance of IBCG (blue) vs SBFW (red) and SBFW with Hessian inverse (green) on Problem (44) when $\mu_g = 0.1$. Plots from left to right are trajectories of $\theta_k$ and $f(\lambda_k, \theta_k) - f^*$.

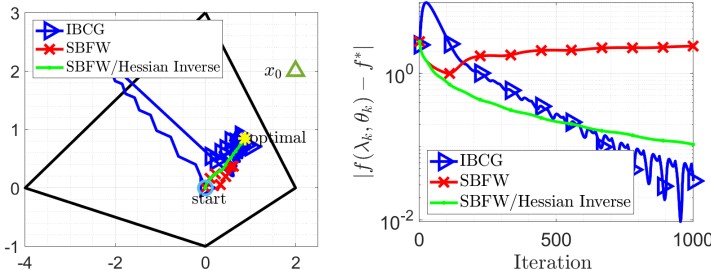

Figure 5: The performance of IBCG (blue) vs SBFW (red) and SBFW with Hessian inverse (green) on Problem (44) when $\mu_g = 0.1$ and the SBFW parameters are tuned. Plots from left to right are trajectories of $\theta_k$ and $f(\lambda_k, \theta_k) - f^*$.

### G.3    MATRIX COMPLETION WITH DENOISING

#### G.3.1    SYNTHETIC DATASET

**Dataset Generation.** We create an observation matrix $M = \hat{X} + E$. In this setting $\hat{X} = WW^T$ where $W \in \mathbb{R}^{n \times r}$ containing normally distributed independent entries, and $E = \hat{n}(L + L^T)$ is a noise matrix where $L \in \mathbb{R}^{n \times n}$ containing normally distributed independent entries and $\hat{n} \in (0, 1)$ is the noise factor. During the simulation process, we set $n = 250$, $r = 10$, and $\alpha = \|\hat{X}\|_*$.

**Initialization.** All the methods start from the same initial point $\mathbf{x}_0$ and $\mathbf{y}_0$ which are generated randomly. We terminate the algorithms either when the maximum number of iterations $K_{\max} = 10^4$ or the maximum time limit $T_{\max} = 2 \times 10^2$ seconds are achieved.

**Implementation Details.** For our method IBCG, we choose the step-sizes as $\gamma = \frac{1}{4\sqrt{K}}$ to avoid instability due to large initial step-sizes, and set $\alpha = 2/(\mu_g + L_g)$ and $\eta = 0.9 \times \frac{1-\beta}{\mu_g}$. We tuned the step-size $\eta_k$ in the SBFW method by multiplying it by a factor of 0.8, and for the TTSA method, we tuned their step-size $\beta$ by multiplying it by a factor of 0.25.

#### G.3.2    REAL DATASET

In order to emphasize the importance of projection-free bilevel algorithms in practical applications, we conducted further experiments using a larger dataset known as MovieLens 1M. This dataset consists of 1 million ratings provided by 6000 individuals for a total of 4000 movies. In Figure 6 the inferior performance of TTSA algorithm in actual computation time, especially when dealing with large datasets becomes more evident. The observed difference can be attributed to the utilization of the projection operation in contrast to the projection-free algorithms. TTSA requires performing projections over nuclear norm at each iteration which is computationally expensive due to the computation of full singular value decomposition. In contrast, projection-free algorithms IBCG and SBFW solve a linear minimization at each iteration, which only requires the computation of singular vectors corresponding to the largest singular value. On the other hand, considering the slow

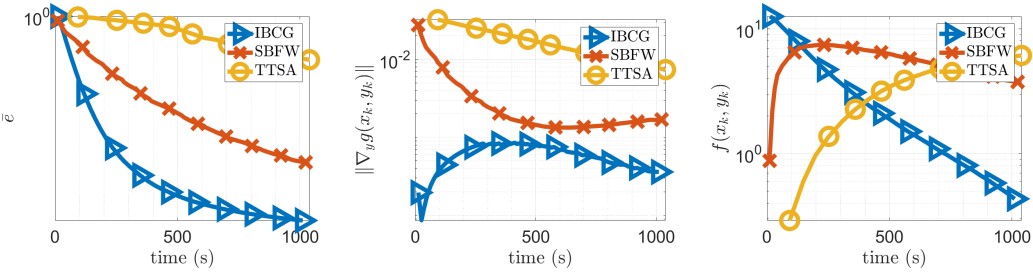

Figure 6: The performance of IBCG (blue) vs SBFW (red) and TTSA (yellow) on Problem (2) for real dataset. Plots from left to right are trajectories of normalized error $(\bar{e})$, $\|\nabla_y g(x_k, y_k)\|$, and $f(x_k, y_k)$ over time.

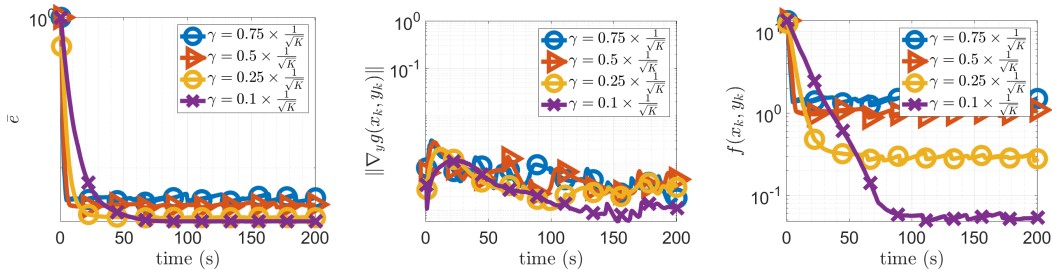

Figure 7: The performance of IBCG on Problem (2) for fixed step-sizes $\eta$ and $\alpha$. Plots from left to right are trajectories of normalized error $(\bar{e})$, $\|\nabla_y g(x_k, y_k)\|$, and $f(x_k, y_k)$ over time.

convergence rate of SBFW, when the size of the dataset increases, the improved performance of our proposed method becomes more evident compared to SBFW.

Moreover, in the following we utilized the MovieLens 100k dataset to implement matrix completion with denoising example for different step-sizes. These experiments will be designed to explore how different step-size selections impact the performance of the IBCG algorithm. We fix the step-sizes $\alpha = 2/(\mu_g + L_g)$ and $\eta = 0.5 \times \frac{1-\beta}{\mu_g}$ and systematically alter $\gamma = c_1 \times \frac{1}{\sqrt{K}}$ with constants $c_1 \in \{0.75, 0.5, 0.25, 0.1\}$ as depicted in Figure 7. We observe that larger values of $\gamma$ directly affect the performance of the algorithm. This observation matches with our theoretical result as demonstrated in Lemma 6. In particular, the error of approximating the lower-level solution and its Jacobian is directly related to the step-size $\gamma$ and larger values of $\gamma$ contributing to larger errors affecting the upper-level objective value.

In Figure 8, we fixed the step-sizes $\alpha = 2/(\mu_g + L_g)$ and $\gamma = 0.1 \times \frac{1}{\sqrt{K}}$ and changed the value of step-size $\eta = c_2 \times \frac{1-\beta}{\mu_g}$ with constants $c_2 \in \{0.75, 0.5, 0.25, 0.1\}$. The performance of the IBCG is robust due to the various values of step-size $\eta$. This indicates that the choice of $\eta$ does not significantly affect the convergence rate, suggesting that the IBCG method is not overly sensitive to this parameter

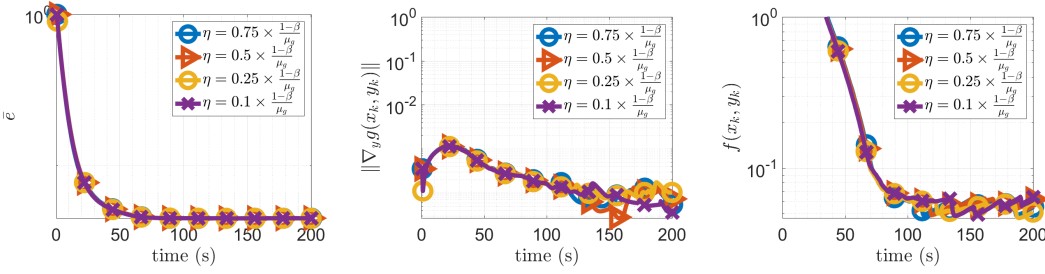

Figure 8: The performance of IBCG on Problem (2) for fixed step-sizes $\gamma$ and $\alpha$. Plots from left to right are trajectories of normalized error $(\bar{e})$, $\|\nabla_y g(x_k, y_k)\|$, and $f(x_k, y_k)$ over time.

within the tested range. The algorithm achieves comparable accuracy levels in the end, regardless of the initial choice of $c_2$, signifying a level of stability that can be beneficial in practical applications where the optimal step-size may not be known at first.

