# OpenReview forum: "An Inexact Conditional Gradient Method for Constrained Bilevel Optimization"
_ICLR.cc/2024/Conference — Submitted to ICLR 2024_

### Official Review · Reviewer_Aef9 · 2023-10-30

**Soundness:** 3 good
**Presentation:** 3 good
**Contribution:** 3 good
**Rating:** 6
**Confidence:** 3

**Summary:**

This paper proposes a novel single-loop projection-free method for the constrained bilevel optimization problem. The paper demonstrates that the method has an improved per-iteration complexity and optimal convergence rate guarantees matching the best-known complexity of projection-free algorithms for solving convex constrained single-level optimization problems. To be specific, the method requires approximately iterations to find an -optimal solution when the upper-level objective function  is convex, and approximately to find an ε-stationary point when  is non-convex.

**Strengths:**

(1) Comparing to the common projection-based methods for constrained bilevel problem, a novel projection-free method with optimal convergence guarantee and lower computational complexity is proposed.

(2) The non-asymptotic optimal convergence rate guarantees are characterized under different settings. And the numerical results are promising.

**Weaknesses:**

The extensive numerical experiments are recommended to conduct.
For example: Since the step-size  is vital to the algorithm’s performance, the numerical experiments of IBCG with respect to the different step-sizes  are advised to conduct as a validation to the theoretical part.

**Questions:**

Theorems 1and 2 give the convergence rate when the hyper-objective is convex and nonconvex. However, it is mentioned in the abstract and the conclusion that the convergence guarantees are achieved under the setting that the upper level objective  is convex and nonconvex. The convexity of  and  is not equivalent. Perhaps I have omitted something. How to make the connection between the function  and  to derive the optimal convergence guarantee when   is convex and nonconvex?

---

> ### Author Response · Authors · 2023-11-20
> **Response to Reviewer Aef9**
>
> **Q1 The algorithm’s performance, the numerical experiments of IBCG with respect to the different step-sizes are advised to conduct as a validation to the theoretical part.**
>
> **A1** Thanks for your suggestion! We have implemented new numerical experiments with different step-sizes to validate our theoretical results, which can be found in Appendix G.3.2 of the revised paper.
>
> ---
>
> **Q2 Theorems 1 and 2 give the convergence rate when $\ell$ is convex and nonconvex. However, it is mentioned in the abstract and the conclusion that the convergence guarantees are achieved under the setting that $f$ is convex and nonconvex.**
>
> **A2** We apologize for the confusion. Indeed, the convexity assumption should be on the hyper-objective function $\ell(x)$, not on the upper-level function $f(x)$. We have revised the abstract and the conclusion sections to clarify this point.

---

### Official Review · Reviewer_Mtbs · 2023-11-01

**Soundness:** 3 good
**Presentation:** 4 excellent
**Contribution:** 2 fair
**Rating:** 5
**Confidence:** 4

**Summary:**

The paper proposes conditional gradient methods for constrained bilevel problems with strongly convex and smooth lower-level problems. The authors develop algorithms with finite time convergence guarantees for solving this class of problems under a deterministic setting. Finally, the authors experimentally evaluate the proposed algorithms to corroborate the theoretical guarantees.

**Strengths:**

Here, I list the main strengths of the paper.

- The paper is well-written and easy to read. All the ideas are clearly presented with sufficient discussions.
- The authors remove the boundedness assumption on $\nabla_y f(x, \cdot)$ which has been used by previous papers to guarantee convergence of bilevel optimization algorithms.
- In addition to the convex case, the authors have presented results for general non-convex settings too.
- The presented analysis captures the dependence on the condition number which is largely ignored in earlier studies.
- Experiments show better performance of IBCG compared to the state-of-the-art.

**Weaknesses:**

Here, I list my **major** concerns:

- My major concern is with the novelty and the contributions of the paper. First of all, the setting considered and the underlying ideas presented in the paper have already appeared in an earlier paper  (Akhtar et al., 2022) where authors have proposed SBFW. In SBFW, the authors consider a constrained stochastic bilevel problem and develop conditional gradient algorithms for solving the problem. Secondly, the authors of (Akhtar et al., 2022) have considered stochastic problems which are certainly more challenging than the deterministic setting considered in this work. I believe although the authors have shown dependence on condition number and removed the boundedness assumption on the partial gradient of $f$ w.r.t. $y$ (by using compactness of the constraint set) the major contributions may not be enough. In addition, the approximation of the product of Hessian inverse and partial gradient by solving a quadratic problem is quite popular in bilevel optimization and has already appeared in the AmIGO algorithm proposed in (Arbel et al ICLR 2022) and other papers.

- Other than matrix completion and toy problems it would be interesting to see the performance of IBCG on large-scale machine learning problems. Will the algorithm be able to compete with stochastic algorithms in that case since the considered algorithm only works for the deterministic setting?


**Minor**

- Define $\Omega_1$ and $\Omega_2$ in equation (2).

- The definition of training and test data before equation (3) is not clear.

- What is $\omega$ in the discussion before Section 3.1.

- Please make sure that $\mathcal{O}$ and  $\tilde{\mathcal{O}}$ are used consistently throughout the paper.

- In the MAML problem are there algorithms that actually impose sparsity as shown in the formulation of equation (3)?

**Questions:**

Please see the weaknesses section above.

---

> ### Author Response · Authors · 2023-11-20
> **Response to Reviewer Mtbs**
>
> First, Thank the reviewer for raising these points, in the following we address your concerns:
>
> **Q1 Novelty of the paper compared to SBFW and AmIGO.**
>
> **A1**  While our IBCG method shares some similarities with the existing methods, such as SBFW and AmIGO, we believe that our paper makes a notable contribution to the study of constrained bilevel optimization. Specifically, our method offers the best-known convergence rate for the considered setting within the class of projection-free algorithms, as shown in Table 1 of the paper. Moreover, to achieve such a result, we also need to address several specific technical challenges, which we highlight below.
>
> - Although some of the techniques used in our analysis may have appeared in prior work, our unique contribution lies in the way we combine and customize these techniques to develop a projection-free method for solving a constrained bilevel optimization problem with a strongly convex lower-level objective function. By tracking lower-level optimal solution trajectories, utilizing a nested approximation involving Hessian inverse information, and employing a projection-free approach, we have proposed a more efficient and streamlined algorithm that performs well in the numerical experiments.
>
> - Specifically, one key distinction of our IBCG method from SBFW and AmIGO is the way we approximate the Hessian matrix inversion in the hyper-gradient in (5). In our approach, we obtain $\tilde{\mathbf{v}}(x_k)$ by performing one step of gradient descent on the quadratic function in (8). This leads to a single-loop algorithm that only requires two matrix-vector products per iteration. On the other hand, SBFW constructs a biased stochastic estimate of the Hessian inverse matrix (as shown in (12) in [R1]) and requires $O(\kappa_g)$ matrix-vector products per iteration. Moreover, AmIGO obtains $\tilde{\mathbf{v}}(x_k)$ by running $T = O(\kappa_g)$ steps of gradient descent (see Theorem 1 in [R2]), which also incurs $O(\kappa_g)$ matrix-vector products per iteration.
> As a result of this simpler algorithmic scheme, our analysis is fundamentally different from those in prior work. For instance, Lemma 6 in our paper is new in the context of projection-free algorithms, where we upper bound the error of approximating $\textbf{v}({\bf x}\_k)$ by ${\bf w}\_{k+1}$.
>
> Due to the space limit in our initial submission, we have focused on tackling the major issues. For the camera-ready version, benefiting from an extra page, we intend to add a related remark on specific technical challenges.
>
> [R1] Akhtar, Zeeshan, et al. ”Projection-free stochastic bi-level optimization.” IEEE Transactions on
> Signal Processing 70 (2022): 6332-6347.
>
> [R2] Arbel, Michael, and Julien Mairal. "Amortized implicit differentiation for stochastic bilevel optimization." arXiv preprint arXiv:2111.14580 (2021).
>
> ---
>
> **Q2 Can compete with stochastic algorithms?**
>
> **A2**  In our experiments, we have found that our proposed algorithm is competitive with stochastic algorithms such as SBFW and TTSA in solving small to medium-scale problems. This may be attributed to the fact that the deterministic nature of our method allows it to explore the search space more systematically and thus converge to high-quality solutions in fewer iterations. On the other hand, we acknowledge that stochastic algorithms have an inherent advantage in handling large-scale problems due to subsampling, which can significantly reduce the computational cost and improve scalability. We would like to explore it in future work.
>
> ---
>
> **Q3 Define $\Omega_1$ and $\Omega_2$.**
>
> **A3** Following your suggestion, we have added the definition of $\Omega_1$ and $\Omega_2$ after equation (2).
>
> ---
>
> **Q4 The definition of training and test data before equation (3) is not clear.**
>
> **A4** Thanks for catching the typo! We have corrected it in the revised paper.
>
> ---
>
> **Q5  What is $\omega$ in the discussion before Section 3.1.**
>
> **A5** Thanks for the question. Here, $\omega$ is the exponent of matrix multiplication in complexity theory, and the current best value is $\omega \approx 2.37$. We have clarified this point in the revised paper.
>
> ---
>
> **Q6  Make sure that $\mathcal{O}$ and $\mathcal{\tilde{O}}$ are used consistently.**
>
> **A6**  We will check the paper to ensure consistency.
>
> ---
>
> **Q7  Other algorithms that  impose sparsity as shown in equation (3)?**
>
>  **A7**  Similar assumption have been proposed in sparse MAML [R1] and section 6.2 of [R2] in which they have considered $\ell_1$ regularization, which is similar to imposing $\ell_1$-norm constraint.
>
> [R1] Gai, S., and Wang, D. (2019, September). Sparse model-agnostic meta-learning algorithm for few-shot learning. In 2019 2nd China Symposium on Cognitive Computing and Hybrid Intelligence (CCHI) (pp. 127-130). IEEE.
>
> [R2] Huang, F., Li, J., Gao, S., and Huang, H. (2022). Enhanced bilevel optimization via bregman distance. Advances in Neural Information Processing Systems, 35, 28928-28939.
>
> ---

---

### Official Review · Reviewer_scGd · 2023-11-03

**Soundness:** 3 good
**Presentation:** 3 good
**Contribution:** 3 good
**Rating:** 8
**Confidence:** 3

**Summary:**

This paper proposes a projection-free bilevel method which leverages the Frank Wolfe and fully single loop hypergradient estimation techniques. The proposed method achieves the best known convergence rate and performs well in practice.

**Strengths:**

1. The theoretical contribution of this paper is solid and well-established. By the dedicated analysis, they analyze the convergence rate of the proposed fully single loop projection-free bilevel method and demonstrate that it achieves the best-known convergence rate.
2. They relax a relatively restrictive assumption -- gradient boundedness -- in bilevel optimization to the optimizing trajectory, which can then be implied by the boundedness of feasible set of upper-level variable and the gradient continuity.

**Weaknesses:**

1. It is good to analyze the convergence in both nonconvex and convex bilevel setting, but it is unclear that when the composite function $l(x)$ can be convex. The only example I come up with is when $f(x,y^*(x))$ is jointly convex and $y^*(x)$ is linear in $x$, which means lower-level objective is quadratic in $y$. It would be better to explain some sufficient condition for $l(x)$ being convex. Also, as the convexity is required for $l(x)$, there are possibly some over-claims in the contribution and conclusion section saying that 'when upper-level objective $f$ is convex, ...'.

**Questions:**

Is it possible to extend the convergence analysis of IBCG to tackle the stochastic bilevel problem like SBFW?

---

> ### Author Response · Authors · 2023-11-20
> **Response to Reviewer scGd**
>
> **Q1  Example when $\ell(x)$ is convex, and over-claims in the contribution.**
>
> **A1**  Thank you for raising this point! The reviewer is correct; we have made revisions to the contribution and conclusion sections to clarify that the convexity assumption is on the hyper-objective function $\ell(\mathbf{x})$, instead of the upper-level function $f(\mathbf{x})$.
>
> Moreover, as the reviewer has mentioned, one sufficient condition for $\ell(\mathbf{x})$ being convex is when $f$ is jointly convex and $y^*(\mathbf{x})$ is linear in $\mathbf{x}$. As another example, we can consider a min-max optimization problem $\min_{\mathbf{x} \in \mathcal{X}} \max_{\mathbf{y} \in \mathbb{R}^m} f(\mathbf{x},\mathbf{y})$, where $f$ is convex in $\bf x$ and strongly-concave in $\bf y$. Note that this can also be reformulated as a bilevel optimization problem by letting $g({\bf x},{\bf y})=-f({\bf x},{\bf y})$, and the hyper-objective $\ell(\mathbf{x}) = \max_{\mathbf{y}} f(\mathbf{x},\mathbf{y})$ is indeed convex. On the other hand, as also noted by [R1], it seems that there are no general sufficient conditions to establish the convexity of $\ell$, and thus it needs to be verified on a case-by-case basis.  Thank you for raising this excellent point and we have added this discussion to the paper.
>
> [R1] Mingyi Hong, Hoi-To Wai, Zhaoran Wang, and Zhuoran Yang. "A two-timescale stochastic algorithm framework for bilevel optimization: Complexity analysis and application to actor-critic." SIAM Journal on Optimization, 2023.
>
> ---
>
> **Q2 Is it possible to extend the convergence analysis of IBCG to tackle the stochastic bilevel problem like SBFW?**
>
> **A2** Thanks for raising this point! Extending the IBCG method to handle stochastic bilevel problems is indeed an interesting avenue for future work. We think the key challenge lies in controlling the approximation errors of $\\|y_k - y^*(x_k)\\|$ and $\\|w_{k+1}-v(x_k)\\|$, since the contraction properties in (22) and (24) would not hold due to stochastic noises. To tackle these issues, some of the techniques used in SBFW might be useful in this regard. As it is beyond the scope of this paper, we leave this for future work.
>
> [R1] Akhtar, Zeeshan, et al. "Projection-free stochastic bi-level optimization." IEEE Transactions on Signal Processing 70 (2022): 6332-6347.

---

### Official Review · Reviewer_dfVp · 2023-11-06

**Soundness:** 3 good
**Presentation:** 3 good
**Contribution:** 3 good
**Rating:** 6
**Confidence:** 4

**Summary:**

The paper studies bilevel optimization where:
- the inner problem is strongly convex,
- the upper problem is smooth (convex or nonconvex) and constrained over a compact and convex set.

Such problem can be handled with projected gradient descent, but the projection operation may be costly.
Instead, the authors propose a single-loop, inexact and projection free method, coined Inexact Bilevel Conditional Gradient, which only takes one step of Frank Wolfe on the lower problem  (Algorithm 1).
A key in the algorithm design is to view the part of the hypergradient involving an Hessian inverse (the costly part), as the solution of a quadratic optimization problem (Eq 8). The solution $v(x_k)$ of this optimization problem is the approximately updated from one iteration $k$ to the next.
The error in this approximate gradient computation is bounded in Lemma 2/Eq9, which in turns sheds light on how to select the parameter $\gamma$ in FW.

Convergence rates are obtained in terms of Frank-Wolfe gap (Eq 4, Corollary 1) for the non convex case, and suboptimality for the convex case (theorem 2).

**Strengths:**

The topic is very relevant, the paper is clearly written and I did not spot any error. Enriching the toolbox of numerical solutions for bilevel optimization is of great importance, given the latter's ubiquity in many fields, from inverse problems to machine learning.

**Weaknesses:**

- The experimental validation could be more complete. In particular, the method is applied only to low rank penalty, the only case (up to my knowledge) where Frank Wolfe shows a real benefit. What are some other possible applications of the method, where it would outperform accelerated proximal gradient? In other words, what are common sets such that FW's linear minimization operator can be computed efficiently, but projection cannot?
- Although it is not proposed by the authors, what is the interest of the noisy matrix completion formulation compared to a bilevel procedure to tune the regularization strength $\lambda$ of a nuclear norm regularized matrix least square problem? ie
$$\min_{\lambda \geq 0} \Vert M_1 \odot(M - Y^\lambda) \Vert^2 \quad \mathrm{s.t.} \quad Y^\lambda = \text{argmin} \Vert{M_2 \odot (M - Y)}^2 + \lambda \Vert Y \Vert_*$$
- Some tricks exist when the upper optimization is constrained (eg, parametrizing the positive regularization hyperparameter as $\lambda = \exp(\mu)$ as in Pedregosa (2016), which may also improve numerical stability). How does the proposed method compare? when should one use a method or the other?
- The provided code is in Matlab, a proprietary software.

**Questions:**

See weaknesses above.


Remarks
- Given the practical focus of the paper, it seems far-fetched to not consider, above Section 3.1, that the cost of matrix matrix multiplication is $m^3$. Algorithms with lower exponents have huge constant terms and are not beneficial for the kind of $m$ that are dealt with in ML.


Cosmetic remarks (no answer needed)
- one can reformulate linear minimization subproblem : missing "the" before "linear"
- In equation below 6c, $x_k$ and $y_k$ should be in bold (twice).
- Page 2 $K$ is not defined in the convergence rate of the Ghadimi and Wang method.
- A reference to the Frank Wolfe gap (the fact that in the convex case it upper bounds the suboptimality, and that still in the convex case it is the duality gap) may help the reader not so familiar with FW in definition 1.
- numbering all equations would make communication (both with reviewers and amongst future readers) easier
- Remark 4.1 has a number format that does not correspond to the rest (Theorem 2, etc)

---

> ### Author Response · Authors · 2023-11-21
> **Response to Reviewer dfVp**
>
> **Q1 Other applications, and what are common sets such that FW’s linear minimization operator can be computed efficiently, but projection cannot?**
>
> **A1** Thanks for raising this point! Regarding first part of your comment, indeed, other than matrix completion problems, the application of our proposed method includes Model-Agnostic Meta-Learning (MAML) [R1] and Kernel Matrix Learning (KML) [R2], to name a few. In MAML, the goal is to develop a flexible model that can effectively adapt to multiple training sets in order to optimize performance for individual tasks -- see section 2.1 in our paper. It is worth noting that saddle point problems can be viewed as a special case of bilevel optimization problems by letting $g({\bf x},{\bf y})=-f({\bf x},{\bf y})$. Therefore, many examples including KML fit into our setting. In this example, the upper-level constraint set is $\mathcal K\triangleq \lbrace {\bf K}\succeq 0\mid \mbox{trace}({\bf K})=r \rbrace$ for some $r>0$. In fact, the projection onto set $\mathcal K$ requires the eigenvalue decomposition of an $n\times n$ matrix, thus incurring an $\mathcal O(n^3)$ complexity per iteration while the Linear Minimization Oracle (LMO) over $\mathcal K$ has a per-iteration complexity $\tilde{\mathcal O}(N_K/\sqrt{\delta})$ where $N_K$ denotes the number of non-zero entries and $\delta$ is the desired accuracy, as elaborated in Table 1 of [R3].
>
> Regarding the second part of your comment, a particularly compelling illustration of the benefits of projection-free techniques is found in the spectrahedron case. Here, rather than needing to calculate a complete singular value decomposition for projections, one can simply compute the principal eigenvector for the LMO, such as through the Lanczos algorithm. There are some sets where the FW algorithm's LMO can be computed efficiently, but projection cannot, or is much harder, including $\ell_p$-norm ball, the nuclear norm ball (also called spectrahedron), matrix trace norm ball,  matrix max-norm ball, Birkhoff polytope and some other examples that can be found in Table 1 of [R3] . Due to the page limit, We have addressed the major issues raised by the reviewers in the revised manuscript. However, we plan to provide a more detailed discussion on this topic in the camera-ready version where we have an extra page.
>
> [R1] Finn, Chelsea, Pieter Abbeel, and Sergey Levine. ”Model-agnostic meta-learning for fast adaptation
> of deep networks.” ICML, 2017.
>
> [R2] Lanckriet, Gert RG, et al. "Learning the kernel matrix with semidefinite programming." Journal of Machine learning research 5.Jan (2004): 27-72.
>
> [R3] Braun, G., Carderera, A., Combettes, C. W., Hassani, H., Karbasi, A., Mokhtari, A., and Pokutta, S. (2022). Conditional gradient methods. arXiv preprint arXiv:2211.14103.
>
> **Q2 What is the interest of the noisy matrix completion formulation compared to a bilevel procedure to tune the regularization strength of $\lambda$ a nuclear norm regularized matrix least square problem?**
>
> **A2** Thank you for highlighting this important consideration.
> The presented formulation by the reviewer is a valid approach for addressing the matrix completion problem, and it shares similarities with the approach outlined in our paper. We've refined this concept by modifying the regularization term to ensure the lower-level problem attains strong convexity, while strategically positioning the nuclear norm within the upper-level problem. Moreover, given that the regularization parameter $\lambda$ is bounded below by zero, the minimum eigenvalue of the lower-level objective function may not be strictly positive, potentially leading to a multiplicity of solutions at the lower level in the formulation presented by the reviewer. This characteristic is crucial since bilevel optimization problems, where the lower-level objective is merely convex, are generally intractable, as shown in [R1]. Therefore, the structure of matrix completion with denoising example provided in the paper aligns well with the framework of our proposed method to be implemented.
>
> [R1] Chen, L., Xu, J., and Zhang, J. (2023). On bilevel optimization without lower-level strong convexity. arXiv preprint arXiv:2301.00712.

---

> ### Author Response · Authors · 2023-11-21
> **Response to Reviewer dfVp**
>
> **Q3 Some tricks exist when the upper optimization is constrained (eg, parametrizing the positive regularization hyperparameter as $\lambda = exp(\mu)$ as in Pedregosa (2016)**
>
> **A3**
> Thanks for your question. The problem considered in Pedregosa (2016) indeed presents an intriguing formulation; however, it is tailored to a specific subclass of problems and does not encompass the broader spectrum of constrained bilevel optimization problems that our research addresses. In particular, in Pedregosa (2016), the following hyper-parameter optimization problem is considered:
>
> $\min_{\lambda \in \mathcal{D}} \text{loss}(\mathcal{S}_{test}, X(\lambda))$
>
> $\hbox {s.t} \quad X(\lambda) \in  arg\min_x \text{loss}(\mathcal{S}_{train}, x)+e^{\lambda}\||x\||^2$
>
> In this framework, if the domain $\mathcal{D}$ is non-compact, the lower-level objective function may lack strong convexity, resulting in multiple solutions. This characteristic is crucial since bilevel optimization problems, where the lower level is merely convex, are generally intractable. The paper assumes a compact $\mathcal{D}$ with a bounded $\lambda \geq \epsilon$, ensuring strong convexity, however, without such compactness assumption convergence to an optimal solution cannot be guaranteed, potentially impacting the algorithm's stability and performance. Moreover, we would like to note that our proposed method can be implemented to the mentioned problem but due to the lack of sophisticated constraints at the upper level, we may find projection-based algorithms perform better.
>
> **Q4 It seems far-fetched to not consider, above Section 3.1, that the cost of matrix multiplication is  $m^3$. Algorithms with lower exponents have huge constant terms and are not beneficial for the kind of $m$ that are dealt with in ML.**
>
> **A4** That is an excellent point. As the reviewer has correctly pointed out, in most ML applications, the cost of matrix multiplication is  $\mathcal O(m^3)$, and the methods that achieve lower complexity of the form $\mathcal{O}(m^{\omega})$ where $ 2.37 \leq \omega < 3$ often ignore the large constant terms in their computational complexity. We will highlight this in the revised manuscript.
>
> **Q5 Cosmetic remarks.**
>
> **A5** Thanks for raising this point! We have
> corrected them in the revised manuscript.

---

### Meta-Review · Area_Chair_yYwR · 2023-12-05

**Metareview:**

The paper introduces a single-loop, projection-free algorithm for handling constrained bilevel optimization problems. It receives mixed opinions (one reviewer was enthuasiast, the others not so much). The reviewer raises significant issues:

The primary concern is about the originality of the method, especially when compared to the work of (Akhtar et al., 2022). The algorithm SBFW studied is similar, the main differences is that the submitted paper propose a deterministic study instead of the stochastic analysis. Looking at the proofs, it is not clear if there was any challenge in performing this stochastic -> deterministic adaptation.

Another less critical point is that the experiments conducted are minimal, mainly concentrating on low-rank penalty scenarios. This narrow focus limits a thorough evaluation of the method's effectiveness in a variety of applications. Another concerns is that it is unclear how the numerical comparaisons were performed.

Given these considerations and taken into account the relative expertise of the different reviewers and my own assessment, my recommendation is to reject the submission.

**Justification For Why Not Higher Score:**

Paper's limited novelty and restricted experimental validation, which is not sufficiently convincing to overlook the similarities with existing methods.

**Justification For Why Not Lower Score:**

N/A

---

### Decision · Program_Chairs · 2024-01-16

Reject